# DeepMed: Semiparametric Causal Mediation Analysis with Debiased Deep Learning

**Siqi Xu**
Department of Statistics and Actuarial Sciences
University of Hong Kong
Hong Kong SAR, China
sqxu@hku.hk

**Lin Liu**[*]
Institute of Natural Sciences, MOE-LSC,
School of Mathematical Sciences, CMA-Shanghai,
and SJTU-Yale Joint Center for Biostatistics and Data Science
Shanghai Jiao Tong University and Shanghai Artificial Intelligence Laboratory
Shanghai, China
linliu@sjtu.edu.cn

**Zhonghua Liu**[*]
Department of Biostatistics
Columbia University
New York, NY, USA
zl2509@cumc.columbia.edu

## Abstract

Causal mediation analysis can unpack the black box of causality and is therefore a powerful tool for disentangling causal pathways in biomedical and social sciences, and also for evaluating machine learning fairness. To reduce bias for estimating Natural Direct and Indirect Effects in mediation analysis, we propose a new method called DeepMed that uses deep neural networks (DNNs) to cross-fit the infinite-dimensional nuisance functions in the efficient influence functions. We obtain novel theoretical results that our DeepMed method (1) can achieve semiparametric efficiency bound without imposing sparsity constraints on the DNN architecture and (2) can adapt to certain low-dimensional structures of the nuisance functions, significantly advancing the existing literature on DNN-based semiparametric causal inference. Extensive synthetic experiments are conducted to support our findings and also expose the gap between theory and practice. As a proof of concept, we apply DeepMed to analyze two real datasets on machine learning fairness and reach conclusions consistent with previous findings.

## 1 Introduction

Tremendous progress has been made in this decade on deploying deep neural networks (DNNs) in real-world problems (Krizhevsky et al., 2012; Wolf et al., 2019; Jumper et al., 2021; Brown et al., 2022). Causal inference is no exception. In semiparametric causal inference, a series of seminal works (Chen et al., 2020; Chernozhukov et al., 2020; Farrell et al., 2021) initiated the investigation of

---

[*]Co-corresponding authors, alphabetical order

36th Conference on Neural Information Processing Systems (NeurIPS 2022).

statistical properties of causal effect estimators when the nuisance functions (the outcome regressions and propensity scores) are estimated by DNNs. However, there are a few limitations in the current literature that need to be addressed before the theoretical results can be used to guide practice:

(1) Most recent works mainly focus on total effect (Chen et al., 2020; Farrell et al., 2021). In many settings, however, more intricate causal parameters are often of greater interests. In biomedical and social sciences, one is often interested in "mediation analysis" to decompose the total effect into direct and indirect effect to unpack the underlying black-box causal mechanism (Baron and Kenny, 1986). More recently, mediation analysis also percolated into machine learning fairness. For instance, in the context of predicting the recidivism risk, Nabi and Shpitser (2018) argued that, for a "fair" algorithm, sensitive features such as race should have no direct effect on the predicted recidivism risk. If such direct effects can be accurately estimated, one can detect the potential unfairness of a machine learning algorithm. We will revisit such applications in Section 5 and Appendix G.

(2) Statistical properties of DNN-based causal estimators in recent works mostly follow from several (recent) results on the convergence rates of DNN-based nonparametric regression estimators (Suzuki, 2019; Schmidt-Hieber, 2020; Tsuji and Suzuki, 2021), with the limitation of relying on *sparse* DNN architectures. The theoretical properties are in turn evaluated by relatively simple synthetic experiments not designed to generate nearly *infinite-dimensional* nuisance functions, a setting considered by almost all the above related works.

The above limitations raise the tantalizing question whether the available statistical guarantees for DNN-based causal inference have practical relevance. In this work, we plan to partially fill these gaps by developing a new method called DeepMed for semiparametric mediation analysis with DNNs. We focus on the *Natural Direct/Indirect Effects* (NDE/NIE) (Robins and Greenland, 1992; Pearl, 2001) (defined in Section 2.1), but our results can also be applied to more general settings; see Remark 2. The DeepMed estimators leverage the "multiply-robust" property of the efficient influence function (EIF) of NDE/NIE (Tchetgen Tchetgen and Shpitser, 2012; Farbmacher et al., 2022) (see Proposition 1 in Section 2.2), together with the flexibility and superior predictive power of DNNs (see Section 3.1 and Algorithm 1). In particular, we also make the following novel contributions to deepen our understanding of DNN-based semiparametric causal inference:

- On the theoretical side, we obtain new results that our DeepMed method can achieve semiparametric efficiency bound without imposing sparsity constraints on the DNN architecture and can adapt to certain low-dimensional structures of the nuisance functions (see Section 3.2), thus significantly advancing the existing literature on DNN-based semiparametric causal inference. *Non-sparse* DNN architecture is more commonly employed in practice (Farrell et al., 2021), and the low-dimensional structures of nuisance functions can help avoid curse-of-dimensionality. These two points, taken together, significantly advance our understanding of the statistical guarantee of DNN-based causal inference.

- More importantly, on the empirical side, in Section 4, we designed sophisticated synthetic experiments to simulate nearly *infinite-dimensional* functions, which are much more complex than those in previous related works (Chen et al., 2020; Farrell et al., 2021; Adcock and Dexter, 2021). We emphasize that these nontrivial experiments could be of independent interest to the theory of deep learning beyond causal inference, to further expose the gap between deep learning theory and practice (Adcock and Dexter, 2021; Gottschling et al., 2020); see Remark 9 for an extended discussion. As a proof of concept, in Section 5 and Appendix G, we also apply DeepMed to re-analyze two real-world datasets on algorithmic fairness and reach similar conclusions to related works.

- Finally, a user-friendly R package can be found at https://github.com/siqixu/DeepMed. Making such resources available helps enhance reproducibility, a highly recognized problem in all scientific disciplines, including (causal) machine learning (Pineau et al., 2021; Kaddour et al., 2022).

## 2 Definition, identification, and estimation of NDE and NIE

### 2.1 Definition of NDE and NIE

Throughout this paper, we denote $Y$ as the primary outcome of interest, $D$ as a binary treatment variable, $M$ as the mediator on the causal pathway from $D$ to $Y$, and $X \in [0,1]^p$ (or more generally,

compactly supported in $\mathbb{R}^p$) as baseline covariates including all potential confounders. We denote the observed data vector as $O \equiv (X, D, M, Y)$. Let $M(d)$ denote the potential outcome for the mediator when setting $D = d$ and $Y(d, m)$ be the potential outcome of $Y$ under $D = d$ and $M = m$, where $d \in \{0, 1\}$ and $m$ is in the support $\mathcal{M}$ of $M$. We define the average total (treatment) effect as $\tau_{tot} := \mathsf{E}[Y(1, M(1)) - Y(0, M(0))]$, the average NDE of the treatment $D$ on the outcome $Y$ when the mediator takes the natural potential outcome when $D = d$ as $\tau_{\mathsf{NDE}}(d) := \mathsf{E}[Y(1, M(d)) - Y(0, M(d))]$, and the average NIE of the treatment $D$ on the outcome $Y$ via the mediator $M$ as $\tau_{\mathsf{NIE}}(d) := \mathsf{E}[Y(d, M(1)) - Y(d, M(0))]$. We have the trivial decomposition $\tau_{tot} \equiv \tau_{\mathsf{NDE}}(d) + \tau_{\mathsf{NIE}}(d')$ for $d \neq d'$. In causal mediation analysis, the parameters of interest are $\tau_{\mathsf{NDE}}(d)$ and $\tau_{\mathsf{NIE}}(d)$.

## 2.2 Semiparametric multiply-robust estimators of NDE/NIE

Estimating $\tau_{\mathsf{NDE}}(d)$ and $\tau_{\mathsf{NIE}}(d)$ can be reduced to estimating $\phi(d, d') := \mathsf{E}[Y(d, M(d'))]$ for $d, d' \in \{0, 1\}$. We make the following standard identification assumptions:

i. **Consistency**: if $D = d$, then $M = M(d)$ for all $d \in \{0, 1\}$; while if $D = d$ and $M = m$, then $Y = Y(d, m)$ for all $d \in \{0, 1\}$ and all $m$ in the support of $M$.

ii. **Ignorability**: $Y(d, m) \perp D | X$, $Y(d, m) \perp M | X, D$, $M(d) \perp D | X$, and $Y(d, m) \perp M(d') | X$, almost surely for all $d, \in \{0, 1\}$ and all $m \in \mathcal{M}$. The first three conditions are, respectively, no unmeasured treatment-outcome, mediator-outcome and treatment-mediator confounding, whereas the fourth condition is often referred to as the "cross-world" condition. We provide more detailed comments on these four conditions in Appendix A.

iii. **Positivity**: The propensity score $a(d|X) \equiv \mathsf{Pr}(D = d | X) \in (c, C)$ for some constants $0 < c \leq C < 1$, almost surely for all $d \in \{0, 1\}$; $f(m|X, d)$, the conditional density (mass) function of $M = m$ (when $M$ is discrete) given $X$ and $D = d$, is strictly bounded between $[\rho, \bar{\rho}]$ for some constants $0 < \rho \leq \bar{\rho} < \infty$ almost surely for all $m$ in $\mathcal{M}$ and all $d \in \{0, 1\}$.

Under the above assumptions, the causal parameter $\phi(d, d')$ for $d, d' \in \{0, 1\}$ can be identified as either of the following three observed-data functionals:

$$
\phi(d, d') \equiv \mathsf{E}\left[\frac{\mathbb{1}\{D = d\}f(M|X, d')Y}{a(d|X)f(M|X, d)}\right] \equiv \mathsf{E}\left[\frac{\mathbb{1}\{D = d'\}}{a(d'|X)}\mu(X, d, M)\right]
$$
$$
\equiv \int \mu(x, d, m)f(m|x, d')p(x)\,\mathrm{d}m\mathrm{d}x, \tag{1}
$$

where $\mathbb{1}\{\cdot\}$ denotes the indicator function, $p(x)$ denotes the marginal density of $X$, and $\mu(x, d, m) := \mathsf{E}[Y|X = x, D = d, M = m]$ is the outcome regression model, for which we also make the following standard boundedness assumption:

iv. $\mu(x, d, m)$ is also strictly bounded between $[-R, R]$ for some constant $R > 0$.

Following the convention in the semiparametric causal inference literature, we call $a, f, \mu$ "nuisance functions". Tchetgen Tchetgen and Shpitser (2012) derived the EIF of $\phi(d, d')$: $\mathsf{EIF}_{d,d'} \equiv \psi_{d,d'}(O) - \phi(d, d')$, where

$$
\psi_{d,d'}(O) = \frac{\mathbb{1}\{D = d\} \cdot f(M|X, d')}{a(d|X) \cdot f(M|X, d)}(Y - \mu(X, d, M))
$$
$$
+ \left(1 - \frac{\mathbb{1}\{D = d'\}}{a(d'|X)}\right)\int_{m \in \mathcal{M}} \mu(X, d, m)f(m|X, d')\mathrm{d}m + \frac{\mathbb{1}\{D = d'\}}{a(d'|X)}\mu(X, d, M). \tag{2}
$$

The nuisance functions $\mu(x, d, m)$, $a(d|x)$ and $f(m|x, d)$ appeared in $\psi_{d,d'}(o)$ are unknown and generally high-dimensional. But with a sample $\mathcal{D} \equiv \{O_j\}_{j=1}^N$ of the observed data, based on $\psi_{d,d'}(o)$, one can construct the following generic sample-splitting multiply-robust estimator of $\phi(d, d')$:

$$
\widetilde{\phi}(d, d') = \frac{1}{n}\sum_{i \in \mathcal{D}_n} \widetilde{\psi}_{d,d'}(O_i), \tag{3}
$$

where $\mathcal{D}_n \equiv \{O_i\}_{i=1}^n$ is a subset of all $N$ data, and $\widetilde{\psi}_{d,d'}(o)$ replaces the unknown nuisance functions $a, f, \mu$ in $\psi_{d,d'}(o)$ by some generic estimators $\widetilde{a}, \widetilde{f}, \widetilde{\mu}$ computed using the remaining $N - n$ *nuisance*

*sample data*, denoted as $\mathcal{D}_\nu$. Cross-fit is then needed to recover the information lost due to sample splitting; see Algorithm 1. It is clear from (2) that $\widetilde{\phi}(d, d')$ is a consistent estimator of $\phi(d, d')$ as long as any two of $\widetilde{a}, \widetilde{f}, \widetilde{\mu}$ are consistent estimators of the corresponding true nuisance functions, hence the name "multiply-robust". Throughout this paper, we take $n \asymp N - n$ and assume:

  v. Any nuisance function estimators are strictly bounded within the respective lower and upper bounds of $a, f, \mu$.

To further ease notation, we define: for any $d \in \{0, 1\}$, $r_{a,d} \coloneqq \left\{ \int \delta_{a,d}(x)^2 \mathrm{d}F(x) \right\}^{1/2}, r_{f,d} \coloneqq \left\{ \int \delta_{f,d}(x, m)^2 \mathrm{d}F(x, m | d = 0) \right\}^{1/2}$, and $r_{\mu,d} \coloneqq \left\{ \int \delta_{\mu,d}(x, m)^2 \mathrm{d}F(x, m | d = 0) \right\}^{1/2}$, where $\delta_{a,d}(x) \coloneqq \widetilde{a}(d | x) - a(d | x), \delta_{f,d}(x, m) \coloneqq \widetilde{f}(m | x, d) - f(m | x, d)$ and $\delta_{\mu,d}(x, m) \coloneqq \widetilde{\mu}(x, d, m) - \mu(x, d, m)$ are point-wise estimation errors of the estimated nuisance functions. In defining the above $L_2$-estimation errors, we choose to take expectation with respect to (w.r.t.) the law $F(m, x | d = 0)$ *only* for convenience, with no loss of generality by Assumptions iii and v.

To show the cross-fit version of $\widetilde{\phi}(d, d')$ is semiparametric efficient for $\phi(d, d')$, we shall demonstrate under what conditions $\sqrt{n}(\widetilde{\phi}(d, d') - \phi(d, d')) \xrightarrow{\mathcal{L}} \mathcal{N}(0, \mathsf{E}[\mathsf{EIF}_{d,d'}^2])$ (Newey, 1990). The following proposition on the statistical properties of $\widetilde{\phi}(d, d')$ is a key step towards this objective.

**Proposition 1.** *Denote* $\mathrm{Bias}(\widetilde{\phi}(d, d')) \coloneqq \mathsf{E}[\widetilde{\phi}(d, d') - \phi(d, d') | \mathcal{D}_\nu]$ *as the bias of* $\widetilde{\phi}(d, d')$ *conditional on the nuisance sample* $\mathcal{D}_\nu$. *Under Assumptions i – v,* $\mathrm{Bias}(\widetilde{\phi}(d, d'))$ *is of second-order:*

$$|\mathrm{Bias}(\widetilde{\phi}(d, d'))| \lesssim \max \left\{ r_{a,d} \cdot r_{f,d}, \max_{d'' \in \{0,1\}} r_{f,d''} \cdot r_{\mu,d}, r_{a,d} \cdot r_{\mu,d} \right\}. \tag{4}$$

*Furthermore, if the RHS of (4) is* $o(n^{-1/2})$, *then*

$$\sqrt{n} \left( \widetilde{\phi}(d, d') - \phi(d, d') \right) = \frac{1}{\sqrt{n}} \sum_{i=1}^{n} (\psi_{d,d'}(O_i) - \phi(d, d')) + o(1) \xrightarrow{d} \mathcal{N} \left( 0, \mathsf{E} \left[ \mathsf{EIF}_{d,d'}^2 \right] \right). \tag{5}$$

Although the above result is a direct consequence of the EIF $\psi_{d,d'}(O)$, we prove Proposition 1 in Appendix B for completeness.

**Remark 2.** *The total effect* $\tau_{tot} = \phi(1, 1) - \phi(0, 0)$ *can be viewed as a special case, for which* $d = d'$ *for* $\phi(d, d')$. *Then* $\mathsf{EIF}_{d,d} \equiv \mathsf{EIF}_d$ *corresponds to the nonparametric EIF of* $\phi(d, d) \equiv \phi(d) \equiv \mathsf{E}[Y(d, M(d))]$:

$$\mathsf{EIF}_d = \psi_d(O) - \phi(d) \text{ with } \psi_d(O) = \frac{\mathbb{1}\{D = d\}}{a(d | X)} Y + \left( 1 - \frac{\mathbb{1}\{D = d\}}{a(d | X)} \right) \mu(X, d),$$

*where* $\mu(x, d) \coloneqq \mathsf{E}[Y | X = x, D = d]$. *Hence all the theoretical results in this paper are applicable to total effect estimation. Our framework can also be applied to all the statistical functionals that satisfy a so-called "mixed-bias" property, characterized recently in Rotnitzky et al. (2021). This class includes the quadratic functional, which is important for uncertainty quantification in machine learning.*

## 3 Estimation and inference of NDE/NIE using DeepMed

We now introduce DeepMed, a method for mediation analysis with nuisance functions estimated by DNNs. By leveraging the second-order bias property of the multiply-robust estimators of NDE/NIE (Proposition 1), we will derive statistical properties of DeepMed in this section. The nuisance function estimators by DNNs are denoted as $\widehat{a}, \widehat{f}, \widehat{\mu}$.

### 3.1 Details on DeepMed

First, we introduce the fully-connected feed-forward neural network with the rectified linear units (ReLU) as the activation function for the hidden layer neurons (FNN-ReLU), which will be used to estimate the nuisance functions. Then, we will introduce an estimation procedure using a $V$-fold

cross-fitting with sample-splitting to avoid the Donsker-type empirical-process assumption on the nuisance functions, which, in general, is violated in high-dimensional setup. Finally, we provide the asymptotic statistical properties of the DNN-based estimators of $\tau_{tot}$, $\tau_{\mathsf{NDE}}(d)$ and $\tau_{\mathsf{NIE}}(d)$.

We denote the ReLU activation function as $\sigma(u) \coloneqq \max(u, 0)$ for any $u \in \mathbb{R}$. Given vectors $x, b$, we denote $\sigma_b(x) \coloneqq \sigma(x - b)$, with $\sigma$ acting on the vector $x - b$ component-wise.

Let $\mathcal{F}_{\mathrm{nn}}$ denote the class of the FNN-ReLU functions

$$\mathcal{F}_{\mathrm{nn}} \coloneqq \left\{ f : \mathbb{R}^p \to \mathbb{R}; f(x) = W^{(L)} \sigma_{b^{(L)}} \circ \cdots \circ W^{(1)} \sigma_{b^{(1)}}(x) \right\},$$

where $\circ$ is the composition operator, $L$ is the number of layers (i.e. depth) of the network, and for $l = 1, \cdots, L$, $W^{(l)}$ is a $K_{l+1} \times K_l$-dimensional weight matrix with $K_l$ being the number of neurons in the $l$-th layer (i.e. width) of the network, with $K_1 = p$ and $K_{L+1} = 1$, and $b^{(l)}$ is a $K_l$-dimensional vector. To avoid notation clutter, we concatenate all the network parameters as $\Theta = (W^{(l)}, b^{(l)}, l = 1, \cdots, L)$ and simply take $K_2 = \cdots = K_L = K$. We also assume $\Theta$ to be bounded: $\|\Theta\|_\infty \leq B$ for some universal constant $B > 0$. We may let the dependence on $L, K, B$ explicit by writing $\mathcal{F}_{nn}$ as $\mathcal{F}_{nn}(L, K, B)$.

DeepMed estimates $\tau_{tot}, \tau_{\mathsf{NDE}}(d), \tau_{\mathsf{NIE}}(d)$ by (3), with the nuisance functions $a, f, \mu$ estimated using $\mathcal{F}_{nn}$ with the $V$-fold cross-fitting strategy, summarized in Algorithm 1 below; also see Farbmacher et al. (2022). DeepMed inputs the observed data $\mathcal{D} \equiv \{O_i\}_{i=1}^N$ and outputs the estimated total effect $\widehat{\tau}_{tot}$, NDE $\widehat{\tau}_{\mathsf{NDE}}(d)$ and NIE $\widehat{\tau}_{\mathsf{NIE}}(d)$, together with their variance estimators $\widehat{\sigma}_{tot}^2$, $\widehat{\sigma}_{\mathsf{NDE}}^2(d)$ and $\widehat{\sigma}_{\mathsf{NIE}}^2(d)$.

---

**Algorithm 1** DeepMed with $V$-fold cross-fitting

---

1: Choose some integer $V$ (usually $V \in \{2, 3, \cdots, 10\}$)
2: Split the $N$ observations into $V$ subsamples $I_v \subset \{1, \cdots, N\} \equiv [N]$ with equal size $n = N/V$;
3: **for** $v = 1, \cdots, V$: **do**
4:     Fit the nuisance functions by DNNs using observations in $[N] \setminus I_v$
5:     Compute the nuisance functions in the subsample $I_v$ using the estimated DNNs in step 4
6:     Obtain $\{\widehat{\psi}_d(O_i), \widehat{\psi}_{d,d'}(O_i)\}_{i \in I_v}$ for the subsample $I_v$ based on (2), respectively, with the nuisance functions replaced by their estimates in step 5
7: **end for**
8: Estimate average potential outcomes by $\widehat{\phi}(d) \coloneqq \frac{1}{N} \sum\limits_{i=1}^N \widehat{\psi}_d(O_i)$, $\widehat{\phi}(d, d') \coloneqq \frac{1}{N} \sum\limits_{i=1}^N \widehat{\psi}_{d,d'}(O_i)$
9: Estimate causal effects by $\widehat{\tau}_{tot}, \widehat{\tau}_{\mathsf{NDE}}(d)$ and $\widehat{\tau}_{\mathsf{NIE}}(d)$ with $\widehat{\phi}(d)$ and $\widehat{\phi}(d, d')$
10: Estimate the variances of $\widehat{\tau}_{tot}, \widehat{\tau}_{\mathsf{NDE}}(d)$ and $\widehat{\tau}_{\mathsf{NIE}}(d)$ by:

$$\widehat{\sigma}_{tot}^2 \coloneqq \frac{1}{N^2} \sum_{i=1}^N (\widehat{\psi}_1(O_i) - \widehat{\psi}_0(O_i))^2 - \frac{1}{N} \widehat{\tau}_{tot}^2; \widehat{\sigma}_{\mathsf{NDE}}^2(d) \coloneqq \frac{1}{N^2} \sum_{i=1}^N (\widehat{\psi}_{1,d}(O_i) - \widehat{\psi}_{0,d}(O_i))^2 - \frac{1}{N} \widehat{\tau}_{\mathsf{NDE}}^2(d);$$

$$\widehat{\sigma}_{\mathsf{NIE}}^2(d) \coloneqq \frac{1}{N^2} \sum_{i=1}^N (\widehat{\psi}_{d,1}(O_i) - \widehat{\psi}_{d,0}(O_i))^2 - \frac{1}{N} \widehat{\tau}_{\mathsf{NIE}}^2(d)$$

**Output:** $\widehat{\tau}_{tot}, \widehat{\tau}_{\mathsf{NDE}}(d), \widehat{\tau}_{\mathsf{NIE}}(d), \widehat{\sigma}_{tot}^2, \widehat{\sigma}_{\mathsf{NDE}}^2(d)$ and $\widehat{\sigma}_{\mathsf{NIE}}^2(d)$

---

**Remark 3** (Continuous or multi-dimensional mediators). *For binary treatment $D$ and continuous or multi-dimensional $M$, to avoid nonparametric/high-dimensional conditional density estimation, we can rewrite $\frac{f(m|x,d')}{a(d|x)f(m|x,d)}$ as $\frac{1-a(d|x,m)}{a(d|x,m)(1-a(d|x))}$ by the Bayes' rule and the integral w.r.t. $f(m|x,d')$ in (2) as $\mathsf{E}[\mu(X, d, M)|X = x, D = d']$. Then we can first estimate $\mu(x, d, m)$ by $\widehat{\mu}(x, d, m)$ and in turn estimate $\mathsf{E}[\mu(X, d, M)|X = x, D = d']$ by regressing $\widehat{\mu}(X, d, M)$ against $(X, D)$ using the FNN-ReLU class. We mainly consider binary $M$ to avoid unnecessary complications; but see Appendix G for an example in which this strategy is used. Finally, the potential incompatibility between models posited for $a(d|x)$ and $a(d|x, m)$ and the joint distribution of $(X, A, M, Y)$ is not of great concern under the semiparametric framework because all nuisance functions are estimated nonparametrically; again, see Appendix G for an extended discussion.*

### 3.2 Statistical properties of DeepMed: Non-sparse DNN architecture and low-dimensional structures of the nuisance functions

According to Proposition 1, to analyze the statistical properties DeepMed, it is sufficient to control the $L_2$-estimation errors of nuisance function estimates $\widehat{a}, \widehat{f}, \widehat{\mu}$ fit by DNNs. To ease presentation,

we first study the theoretical guarantees on the $L_2$-estimation error for a generic nuisance function $g : W \in [0,1]^p \to Z \in \mathbb{R}$, for which we assume:

vi. $Z = g(W) + \xi$, with $\xi$ sub-Gaussian with mean zero and independent of $W$.

Note that when $g$ corresponds to $a, f, \mu$, $(W, Z)$ corresponds to $(X, \mathbb{1}(D = 1)), ((X, D), \mathbb{1}(M = 1))$ and $((X, D, M), Y)$, respectively.

We denote the DNN output from the nuisance sample $\mathcal{D}_\nu$ as $\widehat{g}$. For theoretical results, we consider $\widehat{g}$ as the following empirical risk minimizer (ERM):

$$\widehat{g} := \underset{\bar{g} \in \mathcal{F}_{\mathrm{nn}}(L, K, B)}{\arg\min} \sum_{i \in \mathcal{D}_\nu} (Z_i - \bar{g}(W_i))^2 . \tag{6}$$

To avoid model misspecification, one often assumes $g \in \mathcal{G}$, where $\mathcal{G}$ is some *infinite-dimensional* function space. A common choice is $\mathcal{G} = \mathcal{H}_p(\alpha; C)$, the Hölder ball on the input domain $[0,1]^p$, with smoothness exponent $\alpha$ and radius $C$. Hölder space is one of the most well-studied function spaces in statistics and it is convenient to quantify its complexity by a single smoothness parameter $\alpha$; see Appendix C for a review. It is well-known that estimating Hölder functions suffers from curse-of-dimensionality (Stone, 1982). One remedy is to consider the following generalized Hölder space, by imposing certain low-dimensional structures on $g$:

$$\mathcal{H}_k^\dagger(\alpha; C) := \left\{ g(w) = h(\Gamma w) : h \in \mathcal{H}_k(\alpha; C), \Gamma \in \mathbb{R}^{k \times p} \text{ unknown}, k \leq p \right\} .$$

**Remark 4.** *The above definition contains $g(w) = h(w_I)$, where $I \subset \{1, \cdots, p\}$, as a special case, in which $g$ is assumed to only depend on a subset of the feature vector $w$. One can easily generalize the above definition to additive models $g(w) = \sum_{j=1}^p h_j(w_j)$ where $h_j \in \mathcal{H}_{k_j}(\alpha_j; C_j)$, allowing even more modeling flexibility. To avoid complications, we only consider the above simpler model.*

We can show that the ERM estimator $\widehat{g}$ (6) from the FNN-ReLU class $\mathcal{F}_{\mathrm{nn}}(L, K, B)$ attains the optimal estimation rate over $\mathcal{H}_k^\dagger(\alpha; C)$ up to log factors, by choosing the depth and width appropriately without assuming sparse neural nets.

**Lemma 5.** *Under Assumptions iii – vi, if $g \in \mathcal{H}_k^\dagger(\alpha; C)$ for $k \leq p$, with $LK \asymp n^{\frac{k}{2(k+2\alpha)}}$, we have $\sup_{g \in \mathcal{H}_k^\dagger(\alpha; C)} \left\{ \mathsf{E} \left[ (g(W) - \widehat{g}(W))^2 \right] \right\}^{1/2} \lesssim n^{-\frac{\alpha}{2\alpha+k}} (\log n)^3.$*

Lemma 5, together with Proposition 1, implies the main theoretical result of the paper.

**Theorem 6.** *Under Assumptions i – vi and the following condition on $a, f, \mu$: $a \in \mathcal{H}_k^\dagger(\alpha_a; C), f \in \mathcal{H}_k^\dagger(\alpha_f; C), \mu \in \mathcal{H}_k^\dagger(\alpha_\mu; C)$, with*

$$\min \left\{ \frac{\alpha_a}{2\alpha_a + k} + \frac{\alpha_f}{2\alpha_f + k}, \frac{\alpha_f}{2\alpha_f + k} + \frac{\alpha_\mu}{2\alpha_\mu + k}, \frac{\alpha_a}{2\alpha_a + k} + \frac{\alpha_\mu}{2\alpha_\mu + k} \right\} > \frac{1}{2} + \epsilon, \tag{7}$$

*for $k \leq p$ and some arbitrarily small $\epsilon > 0$, if $\widehat{a}, \widehat{f}, \widehat{\mu}$ are respectively the ERM (6) from FNN-ReLU classes $\mathcal{F}_{nn}(L_a, K_a, B), \mathcal{F}_{nn}(L_f, K_f, B), \mathcal{F}_{nn}(L_\mu, K_\mu, B)$, of which the product of the depth and width satisfies $L_g K_g \asymp n^{\frac{k}{2(k+2\alpha_g)}}$ for $g \in \{a, f, \mu\}$, then the DeepMed estimators $\widehat{\tau}_{tot}, \widehat{\tau}_{\mathsf{NDE}}(d)$ and $\widehat{\tau}_{\mathsf{NIE}}(d)$ computed by Algorithm 1 are semiparametric efficient:*

$$\widehat{\sigma}_{tot}^{-1}(\widehat{\tau}_{tot} - \tau_{tot}), \widehat{\sigma}_{\mathsf{NDE}}^{-1}(d)(\widehat{\tau}_{\mathsf{NDE}}(d) - \tau_{\mathsf{NDE}}(d)), \widehat{\sigma}_{\mathsf{NIE}}^{-1}(d)(\widehat{\tau}_{\mathsf{NIE}}(d) - \tau_{\mathsf{NIE}}(d)) \xrightarrow{\mathcal{L}} \mathcal{N}(0,1), \text{ with}$$

$N\widehat{\sigma}_{tot}^2 \xrightarrow{p} \mathsf{E}[(\mathsf{EIF}_1 - \mathsf{EIF}_0)^2]$, $N\widehat{\sigma}_{\mathsf{NDE}}^2(d) \xrightarrow{p} \mathsf{E}[(\mathsf{EIF}_{1,d} - \mathsf{EIF}_{0,d})^2]$, and $N\widehat{\sigma}_{\mathsf{NIE}}^2(d) \xrightarrow{p} \mathsf{E}[(\mathsf{EIF}_{d,1} - \mathsf{EIF}_{d,0})^2]$, *i.e. $\widehat{\sigma}_{tot}^2, \widehat{\sigma}_{\mathsf{NDE}}^2$ and $\widehat{\sigma}_{\mathsf{NIE}}^2$ are consistent variance estimators.*

**Remark 7.** *To unload notation in the above theorem, consider the special case where the smoothness of all the nuisance functions coincides, i.e. $\alpha_a = \alpha_f = \alpha_\mu = \alpha$. Then Condition (7) reduces to $\alpha > k/2 + \epsilon$ for some arbitrarily small $\epsilon > 0$. For example, if the covariates $X$ have dimension $p = 2$ and no low-dimensional structures are imposed on the nuisance functions (i.e. $k \equiv p$), one needs $\alpha > 1$ to ensure semiparametric efficiency of the DeepMed estimators.*

We emphasize that Lemma 5 and Theorem 6 do not constrain the network sparsity $S$, better reflecting how DNNs are usually used in practice. Theorem 6 advances results on total and decomposition effect

estimation with non-sparse DNNs (Farrell et al., 2021, Theorem 1) in terms of (1) weaker smoothness conditions and (2) adapting to certain low-dimensional structures of the nuisance functions. The proof of Lemma 5 follows from a combination of the improved DNN approximation rate obtained in Lu et al. (2021); Jiao et al. (2021) and standard DNN metric entropy bound (Suzuki, 2019). We prove Lemma 5 and Theorem 6 in Appendix C for completeness. One weakness of Lemma 5 and Theorem 6, as well as in other contemporary works (Chen et al., 2020; Farrell et al., 2021), is the lack of algorithmic/training process considerations (Chen et al., 2022); see Remark 10 and Appendix E for extended discussions.

**Remark 8** (Explicit input-layer regularization). *Training DNNs in practice involves hyperparameter tuning, including the depth $L$ and width $K$ in Theorem 6 and others like epochs. In the synthetic experiments, we consider the nuisance functions only depending on a $k$-subset of $p$-dimensional input. A reasonable heuristic is to add $L_1$-regularization in the input-layer of the DNN. Then the regularization weight $\lambda$ is also a hyperparameter. In practice, we simply use cross-validation to select the hyperparameters that minimize the validation loss. We leave its theoretical justification and the performance of other alternative approaches such as the minimax criterion (Robins et al., 2020; Cui and Tchetgen Tchetgen, 2019) to future works.*

## 4 Synthetic experiments

In this section and Appendix E, we showcase five synthetic experiments. Since ground truth is rarely known in real data, we believe synthetic experiments play an equally, if not more, important role as real data. Before describing the experimental setups, we garner the following key take-home message:

(a) Compared with the other competing methods, DeepMed exhibits better finite-sample performance in most of our experiments;

(b) Cross-validation for DNN hyperparameter tuning works reasonably well in our experiments;

(c) We find DeepMed with explicit regularization in the input layer improves performance (see Table A2) when the true nuisance functions have certain low-dimensional structures in their dependence on the covariates. Farrell et al. (2021) warned against blind explicit regularization in DNNs for total effect estimation. Our observation does not contradict Farrell et al. (2021) as (1) the purpose of the input-layer regularization is not to control the sparsity of the DNN architecture and (2) we do not further regularize hidden layers;

(d) Experimental setups for Cases 3 to 5 generate nuisance functions that are nearly *infinite-dimensional* and close to the boundary of a Hölder ball with a given smoothness exponent (Liu et al., 2020; Li et al., 2005). Thus these synthetic experiments should be better benchmarks than Cases 1 and 2 or settings in other related works such as Farrell et al. (2021). We hope that these highly nontrivial synthetic experiments are helpful to researchers beyond mediation analysis or causal inference. We share the code for generating these functions as a part of the DeepMed package.

We consider a sample with 10,000 i.i.d. observations. The covariates $X = (X_1, ..., X_p)^\top$ are independently drawn from uniform distribution $\text{Uniform}([-1, 1])$. The outcome $Y$, treatment $D$ and mediator $M$ are generated as follows:

$$D \sim \text{Bernoulli}(\mathfrak{s}(d(X))), M \sim 0.2D + m(X) + \mathcal{N}(0, 1), Y \sim 0.2D + M + y(X) + \mathcal{N}(0, 1),$$

where $\mathfrak{s}(x) := (1 + e^{-x})^{-1}$, and we consider the following three cases to generate the nonlinear functions $d(x), m(x)$ and $y(x)$ in the main text:

• Case 1 (simple functions):

$$d(x) = x_1 x_2 + x_3 x_4 x_5 + \sin x_1, m(x) = 4 \sum_{i=1}^{5} \sin 3x_i, y(x) = (x_1 + x_2)^2 + 5 \sin \sum_{i=1}^{5} x_i.$$

• Case 2 (composition of simple functions): we simulate more complex interactions among covariates by composing simple functions as follow:

$$d(x) = d_2 \circ d_1 \circ d_0(x_1, \cdots, x_5), \text{ with } d_0(x_1, \cdots, x_5) = \left( \prod_{i=1}^{2} x_i, \prod_{i=3}^{5} x_i, \prod_{i=1}^{2} \sin x_i, \prod_{i=3}^{5} \sin x_i \right),$$

$$d_1(a_1, \cdots, a_4) = (\sin(a_1 + a_2), \sin a_2, a_3, a_4), \text{ and } d_2(b_1, \cdots, b_4) = 0.5 \sin(b_1 + b_2) + 0.5(b_3 + b_4),$$

$$m(x) = m_1 \circ m_0(x_1, \ldots, x_5), \text{ with } m_0(x_1, \cdots, x_5) = (\sin x_1, \cdots, \sin x_5), m_1(a_1, \cdots, a_5) = 5 \sin \sum_{i=1}^{5} a_i$$

$$\text{and } y(x) = y_2 \circ y_1 \circ y_0(x_1, \cdots, x_5), \text{ with } y_0(x_1, \cdots, x_5) = \left( \sin \sum_{i=1}^{2} x_i, \sin \sum_{i=3}^{5} x_i, \sin \sum_{i=1}^{5} x_i \right),$$

$$y_1(a_1, a_2, a_3) = (\sin(a_1 + a_2), a_3), \text{ and } y_2(b_1, b_2) = 10 \sin(b_1 + b_2).$$

• Case 3 (Hölder functions): we consider more complex nonlinear functions as follows:

$$d(x) = x_1 x_2 + x_3 x_4 x_5 + 0.5\eta(0.2x_1; \alpha), m(x) = \sum_{i=1}^{5} \eta\left(0.5x_i; \alpha\right), y(x) = x_1 x_2 + 3\eta\left(0.2 \sum_{i=1}^{5} x_i; \alpha\right)$$

where $\eta(x; \alpha) = \sum_{j \in J, l \in \mathbb{Z}} 2^{-j(\alpha+0.25)} w_{j,l}(x)$ with $J = \{0, 3, 6, 9, 10, 16\}$ and $w_{j,l}(\cdot)$ is the D6 father wavelet functions dilated at resolution $j$ shifted by $l$. By construction, $\eta(x; \alpha) \in \mathcal{H}_1(\alpha; B)$ for some known constant $B > 0$ following Härdle et al. (1998, Theorem 9.6). Here we set $\alpha = 1.2$ and the intrinsic dimension $k = 1$. Thus we expect the DeepMed estimators are semiparametric efficient. It is indeed the case based on the columns corresponding to Case 3 in Table 1, suggesting that DNNs can be adaptive to certain low-dimensional structures.

**Remark 9.** *The nuisance functions in Cases 3 − 5 (see Appendix E) are less smooth than what have been considered elsewhere, including Farrell et al. (2021), Chen et al. (2020), and even Adcock and Dexter (2021), a paper dedicated to exposing the gap between theoretical approximation rates and DNN practice. These nuisance functions are designed to be near the boundary of a Hölder ball with a given smoothness exponent as we add wavelets at very high resolution in $\eta(x; \alpha)$. This is the assumption under which most of the known statistical properties of DNNs are developed.*

Table 1: The biases, empirical standard errors (SE) and root mean squared errors (RMSE) of the estimated $\tau_{tot}, \tau_{\mathsf{NDE}}(1)$ and $\tau_{\mathsf{NIE}}(1)$, and the coverage probabilities (CP) of their corresponding 95% confidence intervals. $p = 5$ (no irrelevant covariates). The simulation is based on 200 replicates. The full table including $\widehat{\tau}_{\mathsf{NDE}}(0)$ and $\widehat{\tau}_{\mathsf{NIE}}(0)$ can be found in Table A1 in the Appendix.

| | | Case 1 | | | | Case 2 | | | | Case 3 | | | |
| | Method | Bias | SE | RMSE | CP | Bias | SE | RMSE | CP | Bias | SE | RMSE | CP |
|---|---|---|---|---|---|---|---|---|---|---|---|---|---|
| $\tau_{tot}$ | DeepMed | -0.001 | 0.032 | 0.032 | 0.945 | -0.004 | 0.032 | 0.032 | 0.955 | 0.008 | 0.037 | 0.038 | 0.920 |
| | Lasso | 0.192 | 0.089 | 0.212 | 0.460 | -0.304 | 0.116 | 0.325 | 0.215 | 0.346 | 0.079 | 0.355 | 0.010 |
| | RF | 0.067 | 0.042 | 0.079 | 0.775 | -0.078 | 0.056 | 0.096 | 0.950 | -0.009 | 0.042 | 0.043 | 0.985 |
| | GBM | -0.015 | 0.036 | 0.039 | 0.940 | -0.044 | 0.055 | 0.070 | 0.850 | 0.019 | 0.041 | 0.045 | 0.930 |
| | Oracle | -0.001 | 0.029 | 0.029 | 0.955 | -0.003 | 0.029 | 0.029 | 0.925 | -0.001 | 0.032 | 0.032 | 0.935 |
| $\tau_{\mathsf{NDE}}(1)$ | DeepMed | 0.000 | 0.027 | 0.027 | 0.945 | -0.007 | 0.023 | 0.024 | 0.955 | 0.000 | 0.026 | 0.026 | 0.965 |
| | Lasso | 0.130 | 0.043 | 0.137 | 0.220 | -0.375 | 0.059 | 0.380 | 0.000 | 0.226 | 0.064 | 0.235 | 0.050 |
| | RF | 0.048 | 0.029 | 0.056 | 0.700 | -0.188 | 0.044 | 0.193 | 0.005 | 0.030 | 0.038 | 0.048 | 0.980 |
| | GBM | -0.040 | 0.031 | 0.051 | 0.770 | -0.164 | 0.046 | 0.170 | 0.040 | 0.011 | 0.042 | 0.043 | 0.920 |
| | Oracle | 0.000 | 0.022 | 0.022 | 0.945 | -0.002 | 0.020 | 0.020 | 0.985 | 0.001 | 0.022 | 0.022 | 0.955 |
| $\tau_{\mathsf{NIE}}(1)$ | DeepMed | -0.001 | 0.025 | 0.025 | 0.960 | 0.005 | 0.029 | 0.029 | 0.915 | 0.008 | 0.031 | 0.032 | 0.905 |
| | Lasso | 0.058 | 0.077 | 0.096 | 0.875 | 0.069 | 0.094 | 0.117 | 0.905 | 0.120 | 0.045 | 0.128 | 0.220 |
| | RF | 0.066 | 0.037 | 0.076 | 0.665 | 0.108 | 0.059 | 0.123 | 0.860 | -0.045 | 0.038 | 0.059 | 0.765 |
| | GBM | 0.023 | 0.031 | 0.039 | 0.890 | 0.120 | 0.064 | 0.136 | 0.485 | -0.001 | 0.037 | 0.037 | 0.935 |
| | Oracle | -0.001 | 0.020 | 0.020 | 0.975 | 0.000 | 0.021 | 0.021 | 0.930 | -0.002 | 0.022 | 0.022 | 0.920 |

In all the above cases, $\tau_{tot} = 0.4$ and $\tau_{\mathsf{NDE}}(d) = \tau_{\mathsf{NIE}}(d) = 0.2$ for $d \in \{0, 1\}$. We also consider the cases where the total number of covariates $p = 20$ and $100$ but only the first five covariates are relevant to $Y$, $M$ and $D$. All simulation results are based on 200 replicates. The sigmoid function is used in the final layer when the response variable is binary. For comparison, we also use the Lasso,

random forest (RF) and gradient boosted machine (GBM) to estimate the nuisance functions, and use the true nuisance functions (Oracle) as the benchmark. The Lasso is implemented using the R package "hdm" with a data-driven penalty. The DNN, RF and GBM are implemented using the R packages "keras", "randomForest" and "gbm", respectively. We adopt a 3-fold cross-validation to choose the hyperparameters for DNNs (depth $L$, width $K$, $L_1$-regularization parameter $\lambda$ and epochs), RF (number of trees and maximum number of nodes) and GBM (numbers of trees and depth). We use a completely independent sample for the hyperparameter selection. In this paper, we only use one extra dataset to conduct the cross-validation for hyperparameter selection, so our simulation results are conditional on this extra dataset. We use the cross-entropy loss for the binary response and the mean-squared loss for the continuous response. We fix the batch-size as 100 and the other hyperparameters for the other methods are set to the default values in their R packages. See Appendix E for more details.

We compare the performances of different methods in terms of the biases, empirical standard errors (SE) and root mean squared errors (RMSE) of the estimates as well as the coverage probabilities (CP) of their 95% confidence intervals. When $p = k = 5$ (all covariates are relevant or no low-dimensional structures), DeepMed has smaller bias and RMSE than the other competing methods, and is only slightly worse than Oracle. Lasso has the largest bias and poor CP as expected since it does not capture the nonlinearity of the nuisance functions. RF and GBM also have substantial biases, especially in Case 2 with compositions of simple functions. Overall, DeepMed performs better than the competing methods (Table 1). From the empirical distributions, we can also see that they are nearly unbiased and normally distributed in Cases 1-3 (Figures A1-A3). When $p = 20$ or $100$ but only the first five covariates are relevant ($k = 5$), $L_1$-regularization in the input-layer drastically improves the performance of DeepMed (Table A2). DeepMed with $L_1$-regularization in the input-layer also has smaller bias and RMSE than the other competing methods (Tables A3 and A4).

As expected, more precise nuisance function estimates (i.e., smaller validation loss) generally lead to more precise causal effect estimates. The validation losses of nuisance function estimates from DeepMed are generally much smaller than those using Lasso, RF and GBM (Tables A5-A7).

**Remark 10.** *Due to space limitations, we defer Cases 4, 5 to Appendix E, in which* DeepMed *fails to be semiparametric efficient, compared to the Oracle; see an extended discussion in Appendix E. We conjecture this may be due to the implicit regularization of gradient-based training algorithm such as SGD (Table A11) or* adam *(Kingma and Ba, 2015) (all simulation results except Table A11), which is used to train the DNNs to estimate the nuisance parameters, instead of actually solving the ERM* (6). *Most previous works focus on the benefit of implicit regularization (Neyshabur, 2017; Bartlett et al., 2020) on generalization. Yet, implicit regularization might inject implicit bias into causal effect estimates, which could make statistical inference invalid. Such a potential curse of implicit regularization has not been documented in the DNN-based causal inference literature before and exemplify the value of our synthetic experiments. We believe this is an important open research direction for theoretical results to better capture the empirical performance of DNN-based causal inference methods such as* DeepMed.

## 5  Real data analysis on fairness

As a proof of concept, we use DeepMed and other competing methods to re-analyze the COMPAS algorithm (Dressel and Farid, 2018). In particular, we are interested in the NDE of race $D$ on the recidivism risk (or the COMPAS score) $Y$ with the number of prior convictions as the mediator $M$. For race, we mainly focus on the Caucasians population ($D = 0$) and the African-Americans population ($D = 1$), and exclude the individuals of other ethnicity groups. The COMPAS score ($Y$) is ordinal, ranging from 1 to 10 (1: lowest risk; 10: highest risk). We also include the demographic information (age and gender) as covariates $X$.

All the methods find significant positive NDE of race on the COMPAS score at $\alpha$-level 0.005 (Table 2; all p-values $< 10^{-7}$), consistent with previous findings (Nabi and Shpitser, 2018). Thus the COMPAS algorithm tends to assign higher recidivism risks to African-Americans than to Caucasians, even when they have the same number of prior convictions. The validation losses of nuisance function estimates by DeepMed are smaller than the other competing methods (Table A8), possibly suggesting smaller biases of the corresponding NDE/NIE estimators.

We emphasize that research in machine learning fairness should be held accountable (Bao et al., 2021). Our data analysis is merely a proof-of-concept that DeepMed works in practice and the conclusion from our data analysis should not be treated as definitive. We defer the comments on potential issues of unmeasured confounding to Appendix F and another real data analysis to Appendix G.

Table 2: Results for real data application to COMPAS algorithm fairness.

| Method | Effect | Estimate | SE | Method | Effect | Estimate | SE |
|---|---|---|---|---|---|---|---|
| DeepMed | $\tau_{tot}$ | 1.136 | 0.069 | RF | $\tau_{tot}$ | 1.083 | 0.111 |
|  | $\tau_{\text{NDE}}(1)$ | 0.564 | 0.068 |  | $\tau_{\text{NDE}}(1)$ | 0.589 | 0.070 |
|  | $\tau_{\text{NDE}}(0)$ | 0.524 | 0.062 |  | $\tau_{\text{NDE}}(0)$ | 0.569 | 0.103 |
|  | $\tau_{\text{NIE}}(1)$ | 0.612 | 0.042 |  | $\tau_{\text{NIE}}(1)$ | 0.514 | 0.049 |
|  | $\tau_{\text{NIE}}(0)$ | 0.572 | 0.051 |  | $\tau_{\text{NIE}}(0)$ | 0.494 | 0.065 |
| Lasso | $\tau_{tot}$ | 1.150 | 0.068 | GBM | $\tau_{tot}$ | 1.180 | 0.068 |
|  | $\tau_{\text{NDE}}(1)$ | 0.575 | 0.063 |  | $\tau_{\text{NDE}}(1)$ | 0.550 | 0.063 |
|  | $\tau_{\text{NDE}}(0)$ | 0.587 | 0.062 |  | $\tau_{\text{NDE}}(0)$ | 0.526 | 0.061 |
|  | $\tau_{\text{NIE}}(1)$ | 0.563 | 0.032 |  | $\tau_{\text{NIE}}(1)$ | 0.654 | 0.041 |
|  | $\tau_{\text{NIE}}(0)$ | 0.575 | 0.040 |  | $\tau_{\text{NIE}}(0)$ | 0.630 | 0.044 |

## 6 Conclusion and Discussion

In this paper, we proposed DeepMed for semiparametric mediation analysis with DNNs. We established novel statistical properties for DNN-based causal effect estimation that can (1) circumvent sparse DNN architectures and (2) leverage certain low-dimensional structures of the nuisance functions. These results significantly advance our current understanding of DNN-based causal inference including mediation analysis.

Evaluated by our extensive synthetic experiments, DeepMed mostly exhibits improved finite-sample performance over the other competing machine learning methods. But as mentioned in Remark 10, there is still a large gap between statistical guarantees and empirical observations. Therefore an important future direction is to incorporate the training process while investigating the statistical properties to have a deeper theoretical understanding of DNN-based causal inference. It is also of future research interests to enable DeepMed to handle unmeasured confounding and more complex path-specific effects (Malinsky et al., 2019; Miles et al., 2020), and incorporate other hyperparameter tuning strategies that leverage the multiply-robustness property, such as the minimax criterion (Robins et al., 2020; Cui and Tchetgen Tchetgen, 2019).

Finally, we warn readers that all causal inference methods, including DeepMed, may have negative societal impact if they are used without carefully checking their working assumptions.

## Acknowledgement and Disclosure of Funding

The authors thank four anonymous reviewers and one anonymous area chair for helpful comments, Fengnan Gao for some initial discussion on how to incorporate low-dimensional manifold assumptions using DNNs and Ling Guo for discussion on DNN training. The authors would also like to thank Department of Statistics and Actuarial Sciences at The University of Hong Kong for providing high-performance computing servers that supported the numerical experiments in this paper. L. Liu gratefully acknowledges funding support by Natural Science Foundation of China Grant No.12101397 and No.12090024, Pujiang National Lab Grant No. P22KN00524, Natural Science Foundation of Shanghai Grant No.21ZR1431000, Shanghai Science and Technology Commission Grant No.21JC1402900, Shanghai Municipal Science and Technology Major Project No.2021SHZDZX0102, and Shanghai Pujiang Program Research Grant No.20PJ140890.

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
