# Appendix

## A   More comments on the Ignorability conditions

It is well known that NDE/NIE is not nonparametrically identifiable without assuming the four ignorability conditions listed in Assumption ii: for all $d, d' \in \{0, 1\}$ and $m \in \mathcal{M}$

$$\text{No unmeasured treatment-outcome confounding} : Y(d, m) \perp D | X;$$
$$\text{No unmeasured treatment-mediator confounding} : Y(d, m) \perp M | X, D;$$
$$\text{No unmeasured treatment-mediator confounding} : M(d) \perp D | X;$$
$$\text{Cross-world condition} : Y(d, m) \perp M(d') | X.$$

The first three are standard ignorability conditions; but the fourth one involves "cross-world" potential outcomes $Y(d, m)$ and $M(d')$ when $d \neq d'$. The cross-world assumption is often criticized by researchers who are "interventionists" (Robins et al., 2022) because this condition cannot be empirically verified even by conducting randomized trials. To resolve this issue, many other direct/indirect effects are developed that are identifiable without assuming the cross-world condition, e.g. the interventional direct/indirect effect (IDE/IIE) (VanderWeele et al., 2014). We decided to focus on the more standard NDE/NIE in this paper because the identification formulae of NDE and NIE as in (1) are the same as those of IDE and IIE. It is beyond the scope of this paper to discuss the conceptual (dis)advantages of different types of direct/indirect effects for mediation analysis.

## B   The bias of generic sample-splitting multiply-robust estimators of NDE/NIE

In this section, we prove Proposition 1, which is a consequence of the Proposition below.

**Proposition 11.** *Conditional on the nuisance sample data $\mathcal{D}_\nu$, the bias of $\widetilde{\phi}(d, d')$ as an estimator of $\phi(d, d')$ is of the following second-order form:*

$$
\mathsf{E}\left[\widetilde{\phi}(d, d') - \phi(d, d') | \mathcal{D}_\nu\right]
$$
$$
= \mathsf{E}_X\left[\int_{m \in \mathcal{M}} \left(1 - \frac{a(d'|X)}{\widetilde{a}(d'|X)}\right) \left(\frac{\widetilde{f}(m|X, d')}{f(m|X, d')} - 1\right) \widetilde{\mu}(X, d, m) f(m|X, d') \mathrm{d}m\right]
$$
$$
+ \mathsf{E}_X\left[\int_{m \in \mathcal{M}} \left(1 - \frac{f(m|X, d)}{\widetilde{f}(m|X, d)} \frac{\widetilde{f}(m|X, d')}{f(m|X, d')}\right) (\widetilde{\mu}(X, d, m) - \mu(X, d, m)) f(m|X, d') \mathrm{d}m\right]
$$
$$
+ \mathsf{E}_X\left[\int_{m \in \mathcal{M}} \left(1 - \frac{a(d|X)}{\widetilde{a}(d|X)}\right) \frac{\widetilde{f}(m|X, d')}{\widetilde{f}(m|X, d)} (\widetilde{\mu}(X, d, m) - \mu(X, d, m)) f(m|X, d) \mathrm{d}m\right].
$$

*Consequently, one obtains the following upper bound of the bias:*

$$
\mathrm{Bias}(\widetilde{\phi}(d, d')) \equiv \left|\mathsf{E}\left[\widetilde{\phi}(d, d') - \phi(d, d') | \mathcal{D}_\nu\right]\right|
$$
$$
\lesssim r_{a,d} \cdot r_{f,d} + \max_{d'' \in \{0,1\}} r_{f,d''} \cdot r_{\mu,d} + r_{a,d} \cdot r_{\mu,d}.
$$

(8)

*Proof.* The first statement on the bias follows directly from sample-splitting and the form of the EIF $\psi_{d,d'}(o) - \phi(d, d')$. The second statement is obtained by the application of triangle inequality and Cauchy-Schwarz inequality. □

It is worth noting that the upper bound in (8) by Cauchy-Schwarz inequality is by no means the only analysis strategy. For instance, one could also upper bound the bias by Hölder inequality if convergence rates of DNN-based nuisance function estimators are available in general $L_p$-norms beyond $p = 2$. Proposition 1 is a generalization of the results in Robins et al. (2008); Chernozhukov et al. (2018) to mediation analysis.

## C    Hölder functions, their ERM DNN-based estimators and statistical properties

As in the main text, we denote $\mathcal{H}_p(\alpha; C)$ as the Hölder balls of functions from $\mathbb{R}^p$ to $\mathbb{R}$, with smoothness exponent $\alpha$ and radii $C$ (Triebel, 2010; Giné and Nickl, 2016), formally defined below:

$$
\mathcal{H}_p(\alpha; C) := \begin{cases} \left\{ g : [0,1]^p \to \mathbb{R}; \quad \begin{matrix} \max\limits_{m \in \mathbb{Z}_{\geq 0}^p, |m|_1 < \lfloor \alpha \rfloor} \|\partial^m g\|_\infty \leq C \\ \text{and} \max\limits_{m \in \mathbb{Z}_{\geq 0}^p, |m|_1 = \lfloor \alpha \rfloor} \sup\limits_{w,w' \in [0,1]^p, w \neq w'} \dfrac{|\partial^m g(w) - \partial^m g(w')|}{\|w - w'\|_\infty^{\alpha - \lfloor \alpha \rfloor}} \leq C \end{matrix} \right\} & \alpha \geq 1, \\[2em] \left\{ g : [0,1]^p \to \mathbb{R}; \quad \sup\limits_{w,w' \in [0,1]^p, w \neq w'} \dfrac{|g(w) - g(w')|}{\|w - w'\|_\infty^\alpha} \leq C \right\} & 0 < \alpha < 1. \end{cases}
$$

It is well-known (Stone, 1982) that the minimax optimal convergence rate of estimating $g \in \mathcal{H}_p(\alpha; C)$ in $L_2$-norm is $n^{-\frac{\alpha}{2\alpha + p}}$, suffering from curse-of-dimensionality. As mentioned in the main text, one possibility is to consider the function space $\mathcal{H}_k^\dagger(\alpha; C)$ by assuming that the nuisance functions only depend on the covariates $w \in \mathbb{R}^p$ via a $k$-dimensional linear subspace $\Gamma w$, where $\Gamma \in \mathbb{R}^{k \times p}$ and is unknown.

There exist many estimators attaining the optimal rate: e.g. wavelet projection estimators, kernel estimators, etc. In particular, sparse DNN-based estimators have also shown to attain the optimal rate up to a log-factor (Schmidt-Hieber, 2020; Suzuki, 2019). However, since sparse DNNs are computationally demanding to search over $\mathcal{F}_{nn}$, we prefer results that avoid such sparsity constraints. To this end, it is easy to show the following by adapting the proof of Theorem 1.1 of Lu et al. (2021):

**Lemma 12.** *Given $g \in \mathcal{H}_k^\dagger(\alpha; C)$, for large enough depth and width $L, K \in \mathbb{Z}_{>0}$ and some known constant $B > 0$, there exists $\widetilde{g} \in \mathcal{F}_{nn}(L, K, B)$ such that*

$$
\|g - \widetilde{g}\|_\infty \lesssim \left( \frac{L}{\log L} \frac{K}{\log K} \right)^{-2\alpha/k}.
$$

The proof is straightforward by simply taking the parameter of the input layer to be $W^{(1)} = (\Gamma, -\Gamma) \in \mathbb{R}^{2k \times p}$ and $b^{(1)} = 0$ and the second layer parameters chosen appropriately such that the input becomes $\Gamma x$ before the ReLU activation function. The rest of the proof then follows directly by applying Theorem 1.1 of Lu et al. (2021).

Next, we invoke the metric entropy bound of $\mathcal{F}_{nn}(L, K, B)$ established by Lemma 3 of Suzuki (2019):

**Lemma 13** (Metric entropy bound of $\mathcal{F}_{nn}(L, K, B)$)**.** *Denote the covering number (van der Vaart and Wellner, 1996) of $\mathcal{F}_{nn}(L, K, B)$ w.r.t. $L_\infty$-norm as $N(\epsilon, \mathcal{F}_{nn}(L, K, B), \|\cdot\|_\infty)$. Then for any $\epsilon > 0$, for large enough $L, K \in \mathbb{Z}_{>0}$ and $B > 0$, we have*

$$
\log N(\epsilon, \mathcal{F}_{nn}(L, K, B), \|\cdot\|_\infty) \lesssim (LK)^2 \log\left( \frac{LK}{\epsilon} \right).
$$

Combining the above two lemmas, we are now ready to prove Lemma 5.

*Proof of Lemma 5.* Following Lemma 3.2 of Jiao et al. (2021) or standard $M$-estimation and empirical process theory (van der Vaart and Wellner, 1996), under sub-Gaussian assumption of the noise $\xi$, for the ERM estimator $\widehat{g}$ given in (6), we have

$$
\sup_{g \in \mathcal{H}_k^\dagger(\alpha; C)} \mathsf{E}\left[ (\widehat{g}(W) - g(W))^2 \right] \lesssim \frac{(\log n)^2 \log N(1/n, \mathcal{F}_{nn}(L, K, B), \|\cdot\|_\infty)}{n} + \inf_{\widetilde{g} \in \mathcal{F}_{nn}(L, K, B)} \|\widetilde{g} - g\|_\infty^2
$$

$$
\lesssim \frac{(\log n)^3 (LK)^2 \log(LK)}{n} + \left( \frac{L}{\log L} \frac{K}{\log K} \right)^{-4\alpha/k},
$$

where the second inequality follows from Lemma 12 and 13.

Finally, with a simple bias-variance trade-off argument, we can choose $LK \asymp n^{\frac{k}{2(k+2\alpha)}}$ to obtain the desired rate. $\quad\square$

Before proceeding, we make the following remark regarding the optimality of the results in Theorem 6.

**Remark 14.** *We believe the conditions in Theorem 6 are not tight. For example, when the nuisance functions belong to certain Hölder balls, the sufficient and necessary Hölder-type condition for the existence of semiparametric efficient estimator of $\tau_{tot}$ is $\frac{\alpha_a + \alpha_\mu}{2k} > \frac{1}{4}$. The infinite-order U-statistic estimator of Mukherjee et al. (2017) is the only known semiparametric efficient estimator under the above minimal Hölder-type condition. Yet, their estimators require delicate regularity properties of the estimated nuisance functions, which are difficult to verify for DNNs. It is an interesting open theoretical problem how to achieve semiparametric efficiency under minimal Hölder-type condition even simply for $\tau_{tot}$, when the nuisance functions are estimated by DNNs.*

It is possible to generalize Hölder balls in several directions: e.g. assuming different smoothness exponents in different dimensions of the input (Suzuki, 2019) or composing Hölder functions hierarchically to mimick the composition structure of DNNs (Schmidt-Hieber, 2020) (e.g. Case 5 in Appendix E).

*Proof of Theorem 6.* Theorem 6 is an immediate consequence of Lemma 5 and Proposition 1. □

## D   Tables and figures related to the main text

In this section, we collect tables and figures that are related to Cases 1 – 3 of the simulated experiments and real data analysis of the COMPAS dataset, including Table A1 to Table A8 and Figure A1 to Figure A3.

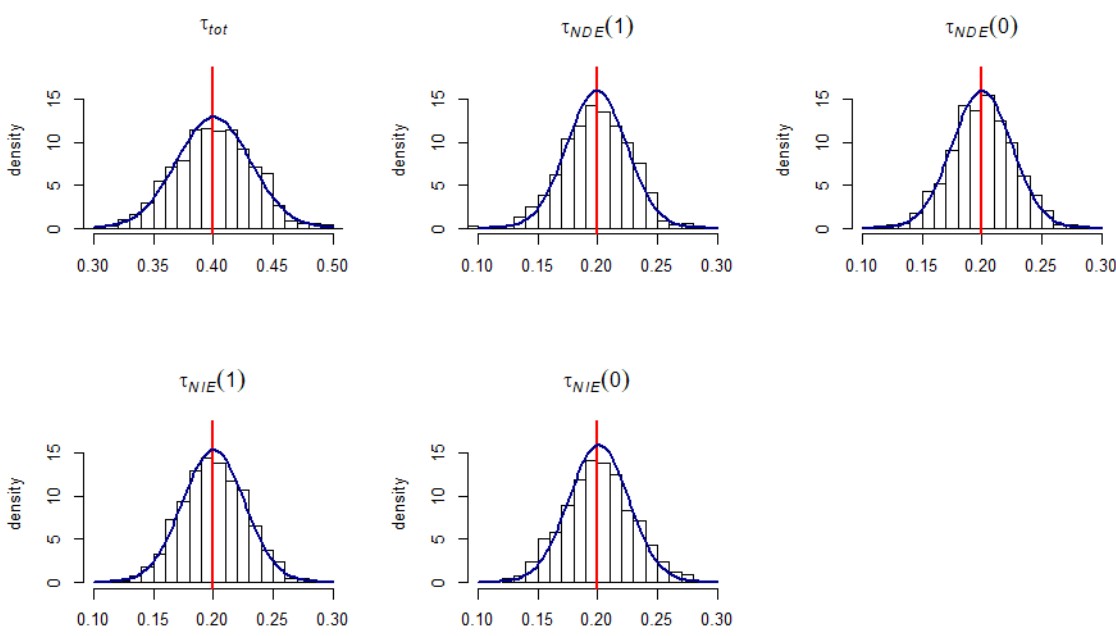

Figure A1: (Case 1) The empirical distributions of the estimated total effects, NIE and NDE by DeepMed. The results are based on 1000 simulation replicates and the number of covariates $p = 5$. The red vertical lines indicate the true effects. The blue curves represent the normal density with the means at the true effects and the estimated standard errors.

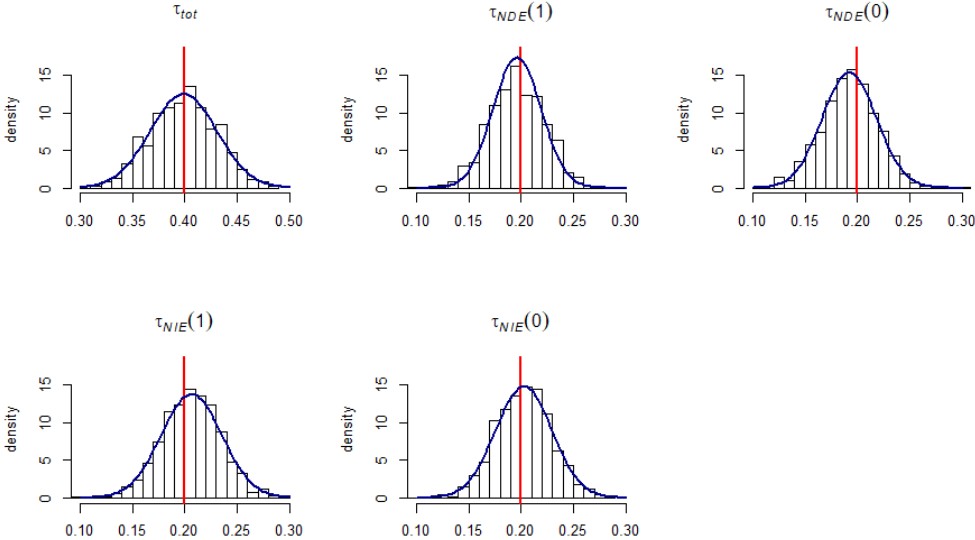

Figure A2: (Case 2) The empirical distributions of the estimated total effects, NIE and NDE by DeepMed. The results are based on 1000 simulation replicates and the number of covariates $p = 5$. The red vertical lines indicate the true effects. The blue curves represent the normal density with the means at the true effects and the estimated standard errors.

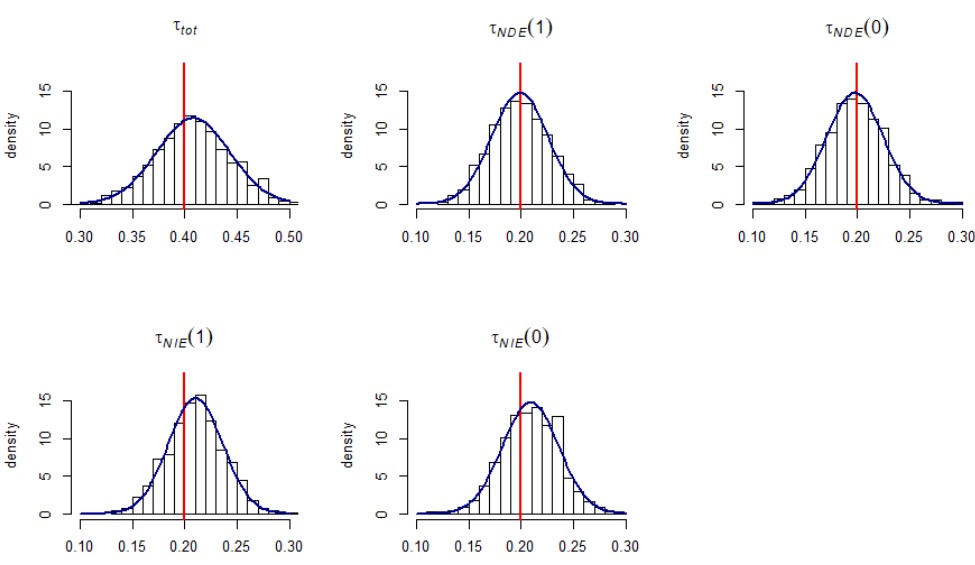

Figure A3: (Case 3) The empirical distributions of the estimated total effects, NIE and NDE by DeepMed. The results are based on 1000 simulation replicates and the number of covariates $p = 5$. The red vertical lines indicate the true effects. The blue curves represent the normal density with the means at the true effects and the estimated standard errors.

Table A1: The biases, empirical standard errors (SE) and root mean squared errors (RMSE) of the estimated average total effects, NDE, and NIE, and the coverage probabilities (CP) of their corresponding 95% confidence intervals. $p = 5$ (no irrelevant covariates). The simulation is based on 200 replicates.

| | | Case 1 | | | | Case 2 | | | | Case 3 | | | |
|---|---|---|---|---|---|---|---|---|---|---|---|---|---|
| | Method | Bias | SE | RMSE | CP | Bias | SE | RMSE | CP | Bias | SE | RMSE | CP |
| $\tau_{tot}$ | DeepMed | -0.001 | 0.032 | 0.032 | 0.945 | -0.004 | 0.032 | 0.032 | 0.955 | 0.008 | 0.037 | 0.038 | 0.920 |
| | Lasso | 0.192 | 0.089 | 0.212 | 0.460 | -0.304 | 0.116 | 0.325 | 0.215 | 0.346 | 0.079 | 0.355 | 0.010 |
| | RF | 0.067 | 0.042 | 0.079 | 0.775 | -0.078 | 0.056 | 0.096 | 0.950 | -0.009 | 0.042 | 0.043 | 0.985 |
| | GBM | -0.015 | 0.036 | 0.039 | 0.940 | -0.044 | 0.055 | 0.070 | 0.850 | 0.019 | 0.041 | 0.045 | 0.930 |
| | Oracle | -0.001 | 0.029 | 0.029 | 0.955 | -0.003 | 0.029 | 0.029 | 0.925 | -0.001 | 0.032 | 0.032 | 0.935 |
| $\tau_{\text{NDE}}(1)$ | DeepMed | 0.000 | 0.027 | 0.027 | 0.945 | -0.007 | 0.023 | 0.024 | 0.955 | 0.000 | 0.026 | 0.026 | 0.965 |
| | Lasso | 0.130 | 0.043 | 0.137 | 0.220 | -0.375 | 0.059 | 0.380 | 0.000 | 0.226 | 0.064 | 0.235 | 0.050 |
| | RF | 0.048 | 0.029 | 0.056 | 0.700 | -0.188 | 0.044 | 0.193 | 0.005 | 0.030 | 0.038 | 0.048 | 0.980 |
| | GBM | -0.040 | 0.031 | 0.051 | 0.770 | -0.164 | 0.046 | 0.170 | 0.040 | 0.011 | 0.042 | 0.043 | 0.920 |
| | Oracle | 0.000 | 0.022 | 0.022 | 0.945 | -0.002 | 0.020 | 0.020 | 0.985 | 0.001 | 0.022 | 0.022 | 0.955 |
| $\tau_{\text{NDE}}(0)$ | DeepMed | 0.000 | 0.025 | 0.025 | 0.940 | -0.009 | 0.026 | 0.028 | 0.915 | -0.001 | 0.027 | 0.027 | 0.940 |
| | Lasso | 0.134 | 0.043 | 0.141 | 0.170 | -0.373 | 0.058 | 0.377 | 0.000 | 0.227 | 0.064 | 0.236 | 0.045 |
| | RF | 0.001 | 0.030 | 0.030 | 0.970 | -0.186 | 0.047 | 0.192 | 0.020 | 0.036 | 0.036 | 0.051 | 0.950 |
| | GBM | -0.037 | 0.030 | 0.048 | 0.800 | -0.164 | 0.049 | 0.171 | 0.055 | 0.020 | 0.044 | 0.048 | 0.920 |
| | Oracle | 0.000 | 0.022 | 0.022 | 0.955 | -0.002 | 0.019 | 0.019 | 0.985 | 0.001 | 0.022 | 0.022 | 0.950 |
| $\tau_{\text{NIE}}(1)$ | DeepMed | -0.001 | 0.025 | 0.025 | 0.960 | 0.005 | 0.029 | 0.029 | 0.915 | 0.008 | 0.031 | 0.032 | 0.905 |
| | Lasso | 0.058 | 0.077 | 0.096 | 0.875 | 0.069 | 0.094 | 0.117 | 0.905 | 0.120 | 0.045 | 0.128 | 0.220 |
| | RF | 0.066 | 0.037 | 0.076 | 0.665 | 0.108 | 0.059 | 0.123 | 0.860 | -0.045 | 0.038 | 0.059 | 0.765 |
| | GBM | 0.023 | 0.031 | 0.039 | 0.890 | 0.120 | 0.064 | 0.136 | 0.485 | -0.001 | 0.037 | 0.037 | 0.935 |
| | Oracle | -0.001 | 0.020 | 0.020 | 0.975 | 0.000 | 0.021 | 0.021 | 0.930 | -0.002 | 0.022 | 0.022 | 0.920 |
| $\tau_{\text{NIE}}(0)$ | DeepMed | -0.001 | 0.028 | 0.028 | 0.940 | 0.003 | 0.027 | 0.027 | 0.930 | 0.008 | 0.029 | 0.030 | 0.910 |
| | Lasso | 0.062 | 0.078 | 0.100 | 0.870 | 0.071 | 0.095 | 0.119 | 0.905 | 0.120 | 0.045 | 0.128 | 0.220 |
| | RF | 0.019 | 0.037 | 0.042 | 0.935 | 0.110 | 0.058 | 0.124 | 0.835 | -0.038 | 0.036 | 0.052 | 0.845 |
| | GBM | 0.025 | 0.031 | 0.040 | 0.910 | 0.120 | 0.059 | 0.134 | 0.495 | 0.008 | 0.037 | 0.038 | 0.935 |
| | Oracle | -0.001 | 0.019 | 0.019 | 0.980 | 0.000 | 0.022 | 0.022 | 0.920 | -0.001 | 0.023 | 0.023 | 0.930 |

# E    Additional information on synthetic experiments

In all the synthetic experiments, we adopt a 3-fold cross-validation to choose the hyperparameters of DNN, RF and GBM over a grid of candidate values. For DNN, we fix the batch-size as 100, and choose depth $L$ from 1 to 3, width $K$ from 10 to 500, $L_1$-regularization parameter $\lambda$ from 0 to 0.4, and epochs from 100 to 500. For RF, we choose the number of trees from 1 to 20, and maximum number of nodes from 10 to 1000. For GBM, we choose the number of trees from 1 to 20, and depth from 10 to 1000. The other hyperparameters are set to the default values in the R packages. As mentioned in the main text, we leave its theoretical justification and other alternative approaches such as the minimax criterion (Robins et al., 2020; Cui and Tchetgen Tchetgen, 2019) or CTMLE (van der Laan and Gruber (2010); also see Chapter 2 of Liu (2018)) to future works.

Case 4 (Hölder functions): we repeat the simulation in Case 3 but set $\alpha = 0.6$ to further decrease the smoothness of the Hölder functions. In particular, $\alpha = 0.6$ is close to the limit ($\alpha = 0.5 + \epsilon$ for arbitrarily small $\epsilon > 0$) for DeepMed estimators to be semiparametric efficient based on Theorem 6. Thus we can examine whether surpassing this limit for nuisance function estimates computed by ERM (6) without considering the DNN training process actually translates to practical success of DeepMed. Unfortunately, the results in Table A9 show otherwise. In general, DeepMed still has superior performance than the other competing methods for $p = 5, 20, 100$. However, even at $p = 5$, the biases of the DeepMed estimators are close to their standard errors. As a result, their CPs undercover the true causal parameters (though the CP is not very far from 95%). However, based on Lemma 5, one should be able to estimate all the nuisance functions at rate $O(n^{-1/4})$ as $\alpha > 0.5$ if the nuisance function estimates are solutions to ERM (6), which should in turn leads to the semiparametric efficiency of DeepMed NDE/NIE estimators and valid inference.

There are several possible explanations for the DeepMed estimators failing to be semiparametric efficient: (1) it is entirely possible that gradient-based training algorithms such as adam (used in our paper) or SGD could find

nuisance function estimators from the FNN-ReLU class that are close to the ERM (6) but we just failed to do so in our implementation; or (2) it is a manifestation of a certain "low-frequency-bias" (Rahaman et al., 2019; Hu et al., 2020; Xu et al., 2020) of DNNs trained by gradient-based algorithms or the computational hardness of learning DNNs by gradient-based algorithms (Goel et al., 2020; Chen et al., 2022), which predicts that DNNs trained by off-the-shelf algorithms bias towards functions with lower complexity. We conjecture that (2) is indeed the reason. It suggests that in future works, to fully establish the practically relevant statistical guarantees of DNN-based nonparametric regression or DNN-based causal inference method such as DeepMed, it is important to take the effect of training algorithms into consideration.

**Remark 15.** *As we have discussed, most established convergence rates of DNNs in the nonparametric statistics literature do not take the potential implicit regularization effect or bias of the training algorithms into account. In this remark, we suggest several possible directions to explore in future works. The so-called Barron space (E et al., 2021) was recently shown to be a natural function space describing the class of neural network functions trained with SGD. If the nuisance functions lie in a Barron space, then the rate of convergence is dimension-independent (E et al., 2021; Chen et al., 2021), which seems to be consistent with the observation that DNNs overcome curse-of-dimensionality in practice. However, as shown in E et al. (2021), the complexity of Barron functions is extremely small, thus casting doubt on if the theoretical claim under Barron spaces is "too good to be true" in fields such as biomedical and social sciences or algorithmic fairness, in which model misspecification bias might have catastrophic consequences. Recent results (Siegel and Xu, 2021) trying to generalize Barron spaces to model more complex functions might be a useful direction to consider in problems related to semiparametric causal inference.*

Case 5 (composition Hölder functions): In the last setting, we consider composition Hölder functions by composing $\eta(x;\alpha) \in \mathcal{H}_1(\alpha; B)$ hierarchically for some constant $B > 0$ as follows:

$$d(x) = 0.2\eta\left(\sum_{i=1}^{3} x_i; \alpha\right) + 0.2\eta\left(\sum_{i=1}^{3} \eta(x_i; \alpha); \alpha\right),$$

$$m(x) = 0.5\eta\left(\sum_{i=1}^{3} x_i; \alpha\right) + 0.2\eta\left(\sum_{i=1}^{3} \eta(x_i; \alpha); \alpha\right),$$

$$y(x) = 0.2\eta\left(\sum_{i=1}^{3} x_i; \alpha\right) + 0.5\eta\left(\sum_{i=1}^{3} \eta(x_i; \alpha); \alpha\right),$$

where we set $\alpha = 1.5$. We choose the "depth" of compositions as 2 for simplicity. The nuisance functions $a, f, \mu$ in Case 5 correspond to the composition Hölder functions studied in the seminal work by Schmidt-Hieber (2020). For such function spaces, Schmidt-Hieber (2020) showed that linear estimators cannot achieve minimax optimal estimation rate, yet nonlinear DNN-based estimators can. As shown in Table A10, the DeepMed estimators do exhibit superior performance compared with the other competing methods but they are far from being semiparametric efficient. When $\alpha = 1.5$, at least based on Schmidt-Hieber (2020), the ERM-based DNN regression estimators should converge to the true function in $L_2$-norm at a rate faster than $n^{-1/4}$. Then using Proposition 1, the DeepMed estimators should have been semiparametric efficient. But as in Table A10, our empirical results suggest otherwise. This is another instance that suggests the necessity of developing more refined theoretical properties of DeepMed.

In results shown previously, adam (Kingma and Ba, 2015) was used to train the DNN weights. Finally, in Table A11, we also display the simulation results when DNN weights were trained by vanilla SGD. Again, as expected, the DeepMed causal effect estimates with DNN weights trained by SGD are not semiparametric efficient.

**Remark 16.** DeepMed *has the option of estimating nuisance functions by other types of machine learning methods, such as those mentioned in Section 4. It is important to develop statistical methodology that can help practitioners decide which method one should use, especially when different methods output qualitatively different results. This is an important research question to pursue as mediation analysis is often followed with critical decision making; a recent proposal can be found in Liu et al. (2020).*

## F   A comment on the potential issue of unmeasured confounding in applications related to algorithmic fairness

Finally, we briefly comment on the potential issue of unmeasured confounding in applications related to algorithmic fairness. It is definitely possible to have unmeasured treatment-mediator confounding in real data analysis. But unmeasured treatment-outcome and mediator-outcome confounding may not be huge issues in mediation analysis

related to algorithmic fairness because in most cases we have access to all the features used to fit the prediction map, which is the outcome $Y$ in our notation. However, they could be violated when some of the features used to fit the model are concealed to protect data privacy. We did not consider these issues in the real data application but we admit that the final results might be biased due to these unmeasured confounding biases. But the same caveat also applies to most of the other works using mediation analysis in algorithmic fairness.

For fields such as epidemiology or social sciences, more often than not, we do not have the luxury of having access to all important features for the mediator, exposure, and outcome. Thus in general, it is important to incorporate instrumental variables (Frölich and Huber, 2017), valid proxies (Dukes et al., 2021) or other identification strategies (Sun and Ye, 2022) into DeepMed to handle unmeasured confounding.

## G   Real data analysis (continued)

This section is a continuation of Section 5 of the main text.

In this section, we apply DeepMed to a second dataset and study whether gender has direct effect on personal annual income not mediated by occupation. We use the Adult dataset (https://archive.ics.uci.edu/ml/datasets/adult) from the 1994 Census database in U.S., which includes 48,842 individuals (Kohavi, 1996). We set $D = 1$ for male and $D = 0$ for female. Occupation ($M$) is a categorical variable containing 14 general types of occupations. The personal annual income is a binary variable, with $Y = 1$ (or $Y = 0$) indicating that an individual makes more (or less) than \$50,000 annually. We also include age, race, education level and employment status as covariates. After removing observations with missing values, the remaining sample size is 45,997.

In this example, since $M$ is multi-dimensional, we utilize the alternative parameterization strategy described in Remark 3 and estimate the propensity scores $a(d|x)$ and $a(d|x, m)$ before and after conditioning on the mediators $M$, together with regressing $\widehat{\mu}(x, d, m)$ against $(x, d)$, all using DNNs. One may be concerned with the potential incoherence between the posited models for the propensity scores $a(d|x, m)$ and $a(d|x)$ and the joint distribution of the observed data $(X, A, M, Y)$. This incoherence could be problematic when parametric models are posited. However, under the semiparametric framework, it is of secondary concern to correctly model the joint distribution of the observed data, which is indeed a very difficult problem. More emphasis is put on how well the target causal parameters such as NDE/NIE are estimated. As long as the nonparametric estimates $\widehat{a}(d|x)$ and $\widehat{a}(d|x, m)$ converge to the true nuisance functions at sufficiently fast rates, the estimates of the target causal parameters should be sufficiently accurate.

All the methods find significant NDE of gender on personal annual income (see Table A12). This positive NDE suggests that males tend to have higher income than females, and this cannot be explained by the indirect effect through occupation. In this dataset, the DeepMed estimators again have smaller validation errors than the other competing methods, possibly suggesting smaller biases of the corresponding NDE/NIE estimators (see Table A13).

# References for Appendix

Sitan Chen, Adam R Klivans, and Raghu Meka. Learning deep ReLU networks is fixed-parameter tractable. In *2021 IEEE 62nd Annual Symposium on Foundations of Computer Science (FOCS)*, pages 696–707. IEEE, 2022.

Ziang Chen, Jianfeng Lu, and Yulong Lu. On the representation of solutions to elliptic PDEs in Barron spaces. *arXiv preprint arXiv:2106.07539*, 2021.

Victor Chernozhukov, Denis Chetverikov, Mert Demirer, Esther Duflo, Christian Hansen, Whitney Newey, and James Robins. Double/debiased machine learning for treatment and structural parameters. *The Econometrics Journal*, 21 (1):C1–C68, 2018.

Yifan Cui and Eric Tchetgen Tchetgen. Selective machine learning of doubly robust functionals. *arXiv preprint arXiv:1911.02029*, 2019.

Oliver Dukes, Ilya Shpitser, and Eric J Tchetgen Tchetgen. Proximal mediation analysis. *arXiv preprint arXiv:2109.11904*, 2021.

Weinan E, Chao Ma, and Lei Wu. The Barron space and the flow-induced function spaces for neural network models. *Constructive Approximation*, pages 1–38, 2021.

Markus Frölich and Martin Huber. Direct and indirect treatment effects–causal chains and mediation analysis with instrumental variables. *Journal of the Royal Statistical Society: Series B (Statistical Methodology)*, 79(5):1645–1666, 2017.

Evarist Giné and Richard Nickl. *Mathematical foundations of infinite-dimensional statistical models*, volume 40. Cambridge University Press, 2016.

Surbhi Goel, Aravind Gollakota, Zhihan Jin, Sushrut Karmalkar, and Adam Klivans. Superpolynomial lower bounds for learning one-layer neural networks using gradient descent. In *International Conference on Machine Learning*, pages 3587–3596. PMLR, 2020.

Wei Hu, Lechao Xiao, Ben Adlam, and Jeffrey Pennington. The surprising simplicity of the early-time learning dynamics of neural networks. *arXiv preprint arXiv:2006.14599*, 2020.

Yuling Jiao, Guohao Shen, Yuanyuan Lin, and Jian Huang. Deep nonparametric regression on approximately low-dimensional manifolds. *arXiv preprint arXiv:2104.06708*, 2021.

Diederik P Kingma and Jimmy Ba. Adam: A method for stochastic optimization. In *International Conference on Learning Representations*, 2015.

Ron Kohavi. Scaling up the accuracy of Naive-Bayes classifiers: a decision-tree hybrid. In *Proceedings of the Second International Conference on Knowledge Discovery and Data Mining*, pages 202–207, 1996.

Lin Liu. *Contributions to Evolutionary Dynamics and Causal Inference*. PhD thesis, 2018.

Lin Liu, Rajarshi Mukherjee, and James M Robins. On nearly assumption-free tests of nominal confidence interval coverage for causal parameters estimated by machine learning. *Statistical Science*, 35(3):518–539, 2020.

Jianfeng Lu, Zuowei Shen, Haizhao Yang, and Shijun Zhang. Deep network approximation for smooth functions. *SIAM Journal on Mathematical Analysis*, 53(5):5465–5506, 2021.

Rajarshi Mukherjee, Whitney K Newey, and James M Robins. Semiparametric efficient empirical higher order influence function estimators. *arXiv preprint arXiv:1705.07577*, 2017.

Nasim Rahaman, Aristide Baratin, Devansh Arpit, Felix Draxler, Min Lin, Fred Hamprecht, Yoshua Bengio, and Aaron Courville. On the spectral bias of neural networks. In *International Conference on Machine Learning*, pages 5301–5310. PMLR, 2019.

James Robins, Lingling Li, Eric Tchetgen Tchetgen, and Aad van der Vaart. Higher order influence functions and minimax estimation of nonlinear functionals. In *Probability and Statistics: Essays in Honor of David A. Freedman*, pages 335–421. Institute of Mathematical Statistics, 2008.

James Robins, Mariela Sued, Quanhong Lei-Gomez, and Andrea Rotnitzky. Double-robust and efficient methods for estimating the causal effects of a binary treatment. *arXiv preprint arXiv:2008.00507*, 2020.

James M Robins, Thomas S Richardson, and Ilya Shpitser. An interventionist approach to mediation analysis. In *Probabilistic and Causal Inference: The Works of Judea Pearl*, pages 713–764. ACM, 2022.

Johannes Schmidt-Hieber. Nonparametric regression using deep neural networks with ReLU activation function. *Annals of Statistics*, 48(4):1875–1897, 2020.

Jonathan W Siegel and Jinchao Xu. Sharp bounds on the approximation rates, metric entropy, and $n$-widths of shallow neural networks. *arXiv preprint arXiv:2101.12365*, 2021.

Charles J Stone. Optimal global rates of convergence for nonparametric regression. *The Annals of Statistics*, 10(4): 1040–1053, 1982.

BaoLuo Sun and Ting Ye. Semiparametric causal mediation analysis with unmeasured mediator-outcome confounding. *Statistica Sinica*, 2022.

Taiji Suzuki. Adaptivity of deep ReLU network for learning in Besov and mixed smooth Besov spaces: optimal rate and curse of dimensionality. In *International Conference on Learning Representations*, 2019.

Hans Triebel. *Theory of Function Spaces*. Modern Birkhäuser Classics. Springer Basel, 2010.

Mark J van der Laan and Susan Gruber. Collaborative double robust targeted maximum likelihood estimation. *The International Journal of Biostatistics*, 6(1), 2010.

Aad W van der Vaart and Jon Wellner. *Weak Convergence and Empirical Processes: with Applications to Statistics*. Springer Science & Business Media, 1996.

Tyler J VanderWeele, Stijn Vansteelandt, and James M Robins. Effect decomposition in the presence of an exposure-induced mediator-outcome confounder. *Epidemiology*, 25(2):300–306, 2014.

Zhi-Qin John Xu, Yaoyu Zhang, Tao Luo, Yanyang Xiao, and Zheng Ma. Frequency principle: Fourier analysis sheds light on deep neural networks. *Communications in Computational Physics*, 28(5):1746–1767, 2020.

Table A2: The biases, empirical standard errors (SE) and root mean squared errors (RMSE) of the total effects, NDE and NIE, and the coverage probabilities (CP) of their corresponding 95% confidence intervals. There exist irrelevant covariates in this setup ($p = 100$). The simulation is based on 200 replicates.

| | | Input-layer $L_1$ regularization | | | | No regularization | | | |
|---|---|---|---|---|---|---|---|---|---|
| | | Bias | SE | RMSE | CP | Bias | SE | RMSE | CP |
| Case 1 | $\tau_{tot}$ | 0.008 | 0.046 | 0.047 | 0.955 | 0.155 | 0.086 | 0.177 | 0.540 |
| | $\tau_{\text{NDE}}(1)$ | 0.009 | 0.043 | 0.044 | 0.920 | 0.061 | 0.044 | 0.075 | 0.745 |
| | $\tau_{\text{NDE}}(0)$ | 0.006 | 0.040 | 0.040 | 0.920 | 0.002 | 0.049 | 0.049 | 0.920 |
| | $\tau_{\text{NIE}}(1)$ | 0.002 | 0.050 | 0.050 | 0.930 | 0.152 | 0.090 | 0.177 | 0.550 |
| | $\tau_{\text{NIE}}(0)$ | -0.001 | 0.044 | 0.044 | 0.945 | 0.094 | 0.081 | 0.124 | 0.800 |
| Case 2 | $\tau_{tot}$ | -0.018 | 0.033 | 0.038 | 0.895 | -0.033 | 0.035 | 0.048 | 0.875 |
| | $\tau_{\text{NDE}}(1)$ | -0.025 | 0.036 | 0.044 | 0.855 | -0.266 | 0.045 | 0.270 | 0.000 |
| | $\tau_{\text{NDE}}(0)$ | -0.018 | 0.036 | 0.040 | 0.855 | -0.306 | 0.051 | 0.310 | 0.000 |
| | $\tau_{\text{NIE}}(1)$ | 0.000 | 0.036 | 0.036 | 0.950 | 0.273 | 0.060 | 0.280 | 0.000 |
| | $\tau_{\text{NIE}}(0)$ | 0.007 | 0.037 | 0.038 | 0.920 | 0.233 | 0.055 | 0.239 | 0.005 |
| Case 3 | $\tau_{tot}$ | 0.019 | 0.043 | 0.047 | 0.915 | 0.224 | 0.055 | 0.231 | 0.015 |
| | $\tau_{\text{NDE}}(1)$ | 0.013 | 0.035 | 0.037 | 0.920 | 0.075 | 0.051 | 0.091 | 0.600 |
| | $\tau_{\text{NDE}}(0)$ | 0.016 | 0.033 | 0.037 | 0.925 | 0.089 | 0.054 | 0.104 | 0.515 |
| | $\tau_{\text{NIE}}(1)$ | 0.003 | 0.038 | 0.038 | 0.940 | 0.135 | 0.064 | 0.149 | 0.340 |
| | $\tau_{\text{NIE}}(0)$ | 0.006 | 0.036 | 0.036 | 0.960 | 0.149 | 0.051 | 0.157 | 0.155 |

Table A3: The biases, empirical standard errors (SE) and root mean squared errors (RMSE) of the estimated total effects, NDE, and NIE, and the coverage probabilities (CP) of their corresponding 95% confidence intervals. There exist irrelevant covariates in this setup ($p = 20$). The simulation is based on 200 replicates.

| | | Case 1 | | | | Case 2 | | | | Case 3 | | | |
|---|---|---|---|---|---|---|---|---|---|---|---|---|---|
| | Method | Bias | SE | RMSE | CP | Bias | SE | RMSE | CP | Bias | SE | RMSE | CP |
| $\tau_{tot}$ | DeepMed | 0.001 | 0.041 | 0.041 | 0.950 | -0.019 | 0.030 | 0.036 | 0.920 | 0.010 | 0.036 | 0.037 | 0.945 |
| | Lasso | 0.191 | 0.091 | 0.212 | 0.510 | -0.318 | 0.108 | 0.336 | 0.155 | 0.334 | 0.077 | 0.343 | 0.010 |
| | RF | 0.035 | 0.049 | 0.060 | 0.955 | -0.187 | 0.076 | 0.202 | 0.475 | 0.100 | 0.052 | 0.113 | 0.545 |
| | GBM | -0.020 | 0.039 | 0.044 | 0.910 | -0.139 | 0.063 | 0.153 | 0.445 | 0.050 | 0.050 | 0.071 | 0.800 |
| | Oracle | -0.002 | 0.032 | 0.032 | 0.925 | -0.003 | 0.029 | 0.029 | 0.925 | -0.003 | 0.030 | 0.030 | 0.950 |
| $\tau_{\text{NDE}}(1)$ | DeepMed | 0.001 | 0.033 | 0.033 | 0.915 | -0.021 | 0.026 | 0.033 | 0.875 | 0.003 | 0.028 | 0.028 | 0.955 |
| | Lasso | 0.129 | 0.048 | 0.138 | 0.205 | -0.378 | 0.060 | 0.383 | 0.000 | 0.216 | 0.062 | 0.225 | 0.065 |
| | RF | 0.033 | 0.037 | 0.050 | 0.890 | -0.213 | 0.043 | 0.217 | 0.000 | 0.088 | 0.047 | 0.100 | 0.545 |
| | GBM | -0.054 | 0.036 | 0.065 | 0.675 | -0.228 | 0.049 | 0.233 | 0.005 | 0.038 | 0.049 | 0.062 | 0.905 |
| | Oracle | -0.002 | 0.023 | 0.023 | 0.925 | -0.002 | 0.020 | 0.020 | 0.985 | -0.002 | 0.020 | 0.020 | 0.970 |
| $\tau_{\text{NDE}}(0)$ | DeepMed | 0.010 | 0.037 | 0.038 | 0.900 | -0.019 | 0.028 | 0.034 | 0.890 | -0.001 | 0.029 | 0.029 | 0.945 |
| | Lasso | 0.133 | 0.048 | 0.141 | 0.185 | -0.376 | 0.060 | 0.381 | 0.000 | 0.216 | 0.064 | 0.225 | 0.075 |
| | RF | 0.007 | 0.038 | 0.039 | 0.955 | -0.212 | 0.044 | 0.217 | 0.000 | 0.081 | 0.046 | 0.093 | 0.605 |
| | GBM | -0.054 | 0.038 | 0.066 | 0.675 | -0.233 | 0.051 | 0.239 | 0.000 | 0.044 | 0.051 | 0.067 | 0.860 |
| | Oracle | -0.002 | 0.023 | 0.023 | 0.930 | -0.002 | 0.019 | 0.019 | 0.985 | -0.002 | 0.020 | 0.020 | 0.960 |
| $\tau_{\text{NIE}}(1)$ | DeepMed | -0.009 | 0.032 | 0.033 | 0.915 | 0.000 | 0.028 | 0.028 | 0.955 | 0.011 | 0.035 | 0.037 | 0.885 |
| | Lasso | 0.059 | 0.078 | 0.098 | 0.890 | 0.058 | 0.093 | 0.110 | 0.920 | 0.118 | 0.042 | 0.125 | 0.270 |
| | RF | 0.028 | 0.040 | 0.049 | 0.965 | 0.025 | 0.079 | 0.083 | 0.965 | 0.019 | 0.033 | 0.038 | 0.895 |
| | GBM | 0.034 | 0.035 | 0.049 | 0.870 | 0.093 | 0.075 | 0.119 | 0.755 | 0.006 | 0.041 | 0.041 | 0.925 |
| | Oracle | 0.000 | 0.021 | 0.021 | 0.950 | 0.000 | 0.021 | 0.021 | 0.930 | 0.000 | 0.023 | 0.023 | 0.935 |
| $\tau_{\text{NIE}}(0)$ | DeepMed | 0.002 | 0.037 | 0.037 | 0.915 | 0.002 | 0.027 | 0.027 | 0.940 | 0.008 | 0.031 | 0.032 | 0.935 |
| | Lasso | 0.062 | 0.079 | 0.100 | 0.870 | 0.060 | 0.094 | 0.112 | 0.915 | 0.119 | 0.042 | 0.126 | 0.220 |
| | RF | 0.002 | 0.039 | 0.039 | 0.980 | 0.025 | 0.078 | 0.082 | 0.965 | 0.012 | 0.032 | 0.034 | 0.920 |
| | GBM | 0.033 | 0.035 | 0.048 | 0.855 | 0.088 | 0.075 | 0.116 | 0.795 | 0.012 | 0.040 | 0.042 | 0.935 |
| | Oracle | 0.000 | 0.021 | 0.021 | 0.940 | 0.000 | 0.022 | 0.022 | 0.920 | -0.001 | 0.023 | 0.023 | 0.925 |

Table A4: The biases, empirical standard errors (SE) and root mean squared errors (RMSE) of the estimated total effects, NDE, and NIE, and the coverage probabilities (CP) of their corresponding $95\%$ confidence intervals. There exist irrelevant covariates in this setup ($p = 100$). The simulation is based on 200 replicates.

| | | Case 1 | | | | Case 2 | | | | Case 3 | | | |
|---|---|---|---|---|---|---|---|---|---|---|---|---|---|
| | Method | Bias | SE | RMSE | CP | Bias | SE | RMSE | CP | Bias | SE | RMSE | CP |
| $\tau_{tot}$ | DeepMed | 0.008 | 0.046 | 0.047 | 0.955 | -0.018 | 0.033 | 0.038 | 0.895 | 0.019 | 0.043 | 0.047 | 0.915 |
| | Lasso | 0.195 | 0.093 | 0.216 | 0.440 | -0.306 | 0.107 | 0.324 | 0.185 | 0.337 | 0.079 | 0.346 | 0.015 |
| | RF | 0.187 | 0.052 | 0.194 | 0.095 | -0.205 | 0.095 | 0.226 | 0.560 | 0.206 | 0.057 | 0.214 | 0.040 |
| | GBM | -0.016 | 0.035 | 0.038 | 0.965 | -0.155 | 0.072 | 0.171 | 0.440 | 0.082 | 0.053 | 0.098 | 0.620 |
| | Oracle | 0.000 | 0.030 | 0.030 | 0.945 | 0.000 | 0.030 | 0.030 | 0.960 | 0.001 | 0.031 | 0.031 | 0.935 |
| $\tau_{NDE}(1)$ | DeepMed | 0.009 | 0.043 | 0.044 | 0.920 | -0.025 | 0.036 | 0.044 | 0.855 | 0.013 | 0.035 | 0.037 | 0.920 |
| | Lasso | 0.129 | 0.050 | 0.138 | 0.275 | -0.369 | 0.055 | 0.373 | 0.000 | 0.215 | 0.067 | 0.225 | 0.095 |
| | RF | 0.104 | 0.040 | 0.111 | 0.305 | -0.216 | 0.048 | 0.221 | 0.000 | 0.155 | 0.054 | 0.164 | 0.150 |
| | GBM | -0.022 | 0.038 | 0.044 | 0.925 | -0.235 | 0.055 | 0.241 | 0.010 | 0.051 | 0.056 | 0.076 | 0.790 |
| | Oracle | 0.001 | 0.022 | 0.022 | 0.950 | 0.000 | 0.021 | 0.021 | 0.935 | 0.002 | 0.022 | 0.022 | 0.930 |
| $\tau_{NDE}(0)$ | DeepMed | 0.006 | 0.040 | 0.040 | 0.920 | -0.018 | 0.036 | 0.040 | 0.855 | 0.016 | 0.033 | 0.037 | 0.925 |
| | Lasso | 0.132 | 0.050 | 0.141 | 0.270 | -0.368 | 0.056 | 0.372 | 0.000 | 0.215 | 0.067 | 0.225 | 0.085 |
| | RF | 0.063 | 0.038 | 0.074 | 0.690 | -0.213 | 0.047 | 0.218 | 0.005 | 0.139 | 0.056 | 0.150 | 0.255 |
| | GBM | -0.032 | 0.037 | 0.049 | 0.870 | -0.241 | 0.054 | 0.247 | 0.005 | 0.059 | 0.057 | 0.082 | 0.765 |
| | Oracle | 0.001 | 0.022 | 0.022 | 0.940 | 0.000 | 0.021 | 0.021 | 0.935 | 0.002 | 0.022 | 0.022 | 0.940 |
| $\tau_{NIE}(1)$ | DeepMed | 0.002 | 0.050 | 0.050 | 0.930 | 0.000 | 0.036 | 0.036 | 0.950 | 0.003 | 0.038 | 0.038 | 0.940 |
| | Lasso | 0.063 | 0.080 | 0.102 | 0.885 | 0.062 | 0.089 | 0.108 | 0.915 | 0.122 | 0.044 | 0.130 | 0.200 |
| | RF | 0.124 | 0.050 | 0.134 | 0.335 | 0.008 | 0.099 | 0.099 | 0.945 | 0.068 | 0.036 | 0.077 | 0.395 |
| | GBM | 0.016 | 0.040 | 0.043 | 0.905 | 0.086 | 0.084 | 0.120 | 0.835 | 0.023 | 0.036 | 0.043 | 0.905 |
| | Oracle | -0.001 | 0.021 | 0.021 | 0.930 | 0.000 | 0.021 | 0.021 | 0.950 | -0.001 | 0.021 | 0.021 | 0.950 |
| $\tau_{NIE}(0)$ | DeepMed | -0.001 | 0.044 | 0.044 | 0.945 | 0.007 | 0.037 | 0.038 | 0.920 | 0.006 | 0.036 | 0.036 | 0.960 |
| | Lasso | 0.066 | 0.081 | 0.104 | 0.870 | 0.063 | 0.090 | 0.110 | 0.915 | 0.123 | 0.044 | 0.131 | 0.215 |
| | RF | 0.084 | 0.047 | 0.096 | 0.645 | 0.011 | 0.098 | 0.099 | 0.945 | 0.051 | 0.034 | 0.061 | 0.615 |
| | GBM | 0.007 | 0.038 | 0.039 | 0.940 | 0.080 | 0.082 | 0.115 | 0.845 | 0.031 | 0.034 | 0.046 | 0.890 |
| | Oracle | -0.001 | 0.021 | 0.021 | 0.935 | 0.000 | 0.021 | 0.021 | 0.935 | -0.001 | 0.021 | 0.021 | 0.960 |

Table A5: The validation loss of the nuisance functions. The cross-entropy loss is used for fitting $a(d|x, m)$ and $a(d|x)$, and mean squared loss is used for fitting the other nuisance functions. There exist no irrelevant covariates in this setup.

| | | $a(1|x, m)$ | $a(1|x)$ | $\mu(x, 1, m)$ | $E_0(\mu_1)^*$ | $\mu(x, 1)$ | $\mu(x, 0, m)$ | $E_1(\mu_0)^*$ | $\mu(x, 0)$ |
|---|---|---|---|---|---|---|---|---|---|
| Case 1 | DeepMed | 0.646 | 0.647 | 1.151 | 1.353 | 2.290 | 1.172 | 1.275 | 2.304 |
| | Lasso | 0.660 | 0.660 | 5.677 | 14.889 | 20.725 | 5.634 | 15.099 | 20.705 |
| | RF | 0.662 | 0.664 | 3.284 | 6.322 | 6.189 | 3.777 | 5.618 | 6.909 |
| | GBM | 0.651 | 0.651 | 2.344 | 3.003 | 3.290 | 2.383 | 2.819 | 3.370 |
| | Oracle | 0.639 | 0.642 | 1.057 | 1.031 | 2.100 | 1.063 | 1.033 | 2.108 |
| Case 2 | DeepMed | 0.680 | 0.681 | 1.309 | 1.434 | 2.318 | 1.305 | 1.213 | 2.311 |
| | Lasso | 0.694 | 0.694 | 8.285 | 23.037 | 31.239 | 8.275 | 23.117 | 31.278 |
| | RF | 0.694 | 0.697 | 5.046 | 19.915 | 16.393 | 4.924 | 19.291 | 16.568 |
| | GBM | 0.688 | 0.689 | 4.587 | 11.961 | 8.089 | 4.441 | 11.592 | 7.918 |
| | Oracle | 0.670 | 0.676 | 1.055 | 1.037 | 2.109 | 1.061 | 1.039 | 2.116 |
| Case 3 | DeepMed | 0.647 | 0.649 | 1.35 | 1.388 | 2.572 | 1.376 | 1.36 | 2.568 |
| | Lasso | 0.662 | 0.664 | 9.31 | 4.612 | 13.935 | 8.945 | 4.584 | 13.35 |
| | RF | 0.657 | 0.664 | 4.966 | 2.728 | 5.388 | 5.101 | 2.647 | 5.425 |
| | GBM | 0.648 | 0.649 | 4.087 | 3.136 | 4.162 | 4.131 | 2.946 | 4.172 |
| | Oracle | 0.637 | 0.643 | 1.031 | 1.019 | 2.11 | 1.033 | 1.02 | 2.111 |

$^*E_0(\mu_1) = E[\mu(X, D = 1, M)|X, D = 0]$ and $E_1(\mu_0) = E[\mu(X, D = 0, M)|X, D = 1]$.

Table A6: The validation loss of the nuisance functions. The cross-entropy loss is used for fitting $a(d|x,m)$ and $a(d|x)$, and mean squared loss is used for fitting the other nuisance functions. There exist irrelevant covariates in this setup ($p = 20$).

| | | $a(1|x,m)$ | $a(1|x)$ | $\mu(x,1,m)$ | $\mathsf{E}_0(\mu_1)^*$ | $\mu(1,x)$ | $\mu(x,0,m)$ | $\mathsf{E}_1(\mu_0)^*$ | $\mu(0,x)$ |
|---|---|---|---|---|---|---|---|---|---|
| Case 1 | DeepMed | 0.657 | 0.659 | 1.316 | 2.943 | 3.241 | 1.454 | 2.709 | 3.319 |
| | Lasso | 0.666 | 0.667 | 5.583 | 14.84 | 20.663 | 5.541 | 15.25 | 20.65 |
| | RF | 0.662 | 0.661 | 4.449 | 8.407 | 8.078 | 4.958 | 7.619 | 8.628 |
| | GBM | 0.657 | 0.658 | 2.987 | 3.24 | 3.976 | 3.11 | 3.265 | 3.992 |
| | Oracle | 0.64 | 0.645 | 1.014 | 1.019 | 2.097 | 1.009 | 1.02 | 2.087 |
| Case 2 | DeepMed | 0.692 | 0.691 | 1.42 | 1.822 | 2.432 | 1.543 | 1.793 | 2.533 |
| | Lasso | 0.694 | 0.694 | 7.98 | 22.882 | 31.023 | 8.003 | 23.34 | 30.977 |
| | RF | 0.692 | 0.692 | 5.708 | 25.493 | 22.644 | 5.618 | 26.092 | 21.803 |
| | GBM | 0.693 | 0.693 | 5.176 | 17.787 | 12.627 | 5.187 | 18.722 | 12.227 |
| | Oracle | 0.672 | 0.677 | 1.014 | 1.017 | 2.095 | 1.01 | 1.018 | 2.083 |
| Case 3 | DeepMed | 0.653 | 0.652 | 1.448 | 2.456 | 3.214 | 1.504 | 2.512 | 3.172 |
| | Lasso | 0.655 | 0.657 | 9.581 | 4.161 | 13.785 | 8.699 | 4.663 | 13.162 |
| | RF | 0.649 | 0.651 | 6.626 | 2.824 | 6.42 | 6.789 | 2.924 | 6.591 |
| | GBM | 0.646 | 0.647 | 6.344 | 3.372 | 5.14 | 6.49 | 3.619 | 5.426 |
| | Oracle | 0.635 | 0.638 | 1.021 | 1.034 | 2.059 | 1.025 | 1.037 | 2.061 |

$^*\mathsf{E}_0(\mu_1) = \mathsf{E}[\mu(X, D = 1, M)|X, D = 0]$ and $\mathsf{E}_1(\mu_0) = \mathsf{E}[\mu(X, D = 0, M)|X, D = 1]$.

Table A7: The validation loss of the nuisance functions. The cross-entropy loss is used for fitting $a(d|x,m)$ and $a(d|x)$, and mean squared loss is used for fitting the other nuisance functions. There exist irrelevant covariates in this setup ($p = 100$).

| | | $a(1|x,m)$ | $a(1|x)$ | $\mu(x,1,m)$ | $\mathsf{E}_0(\mu_1)^*$ | $\mu(x,1)$ | $\mu(x,0,m)$ | $\mathsf{E}_1(\mu_0)^*$ | $\mu(x,0)$ |
|---|---|---|---|---|---|---|---|---|---|
| Case 1 | DeepMed | 0.672 | 0.672 | 1.769 | 4.154 | 4.071 | 1.692 | 4.723 | 4.138 |
| | Lasso | 0.667 | 0.667 | 5.438 | 14.828 | 20.607 | 5.478 | 15.592 | 20.607 |
| | RF | 0.667 | 0.668 | 4.964 | 9.123 | 8.836 | 5.487 | 8.794 | 9.530 |
| | GBM | 0.664 | 0.663 | 3.266 | 3.659 | 4.112 | 3.469 | 3.684 | 4.147 |
| | Oracle | 0.648 | 0.652 | 1.001 | 1.025 | 2.069 | 1.001 | 1.027 | 2.066 |
| Case 2 | DeepMed | 0.695 | 0.694 | 1.81 | 2.583 | 2.909 | 1.926 | 2.615 | 2.936 |
| | Lasso | 0.695 | 0.695 | 8.006 | 22.765 | 31.272 | 8.005 | 23.63 | 31.239 |
| | RF | 0.692 | 0.692 | 5.708 | 25.493 | 22.644 | 5.618 | 26.092 | 21.803 |
| | GBM | 0.688 | 0.689 | 4.587 | 11.961 | 8.089 | 4.441 | 11.592 | 7.918 |
| | Oracle | 0.677 | 0.68 | 1.003 | 1.022 | 2.067 | 1.003 | 1.023 | 2.065 |
| Case 3 | DeepMed | 0.679 | 0.677 | 2.416 | 4.563 | 4.643 | 2.500 | 5.098 | 4.621 |
| | Lasso | 0.660 | 0.662 | 9.784 | 4.858 | 14.364 | 8.523 | 4.360 | 13.000 |
| | RF | 0.661 | 0.663 | 8.124 | 3.102 | 7.808 | 7.973 | 2.970 | 7.586 |
| | GBM | 0.653 | 0.653 | 7.906 | 3.320 | 6.205 | 7.252 | 3.101 | 6.159 |
| | Oracle | 0.635 | 0.641 | 1.011 | 1.039 | 2.144 | 1.014 | 1.042 | 2.144 |

$^*\mathsf{E}_0(\mu_1) = \mathsf{E}[\mu(X, D = 1, M)|X, D = 0]$ and $\mathsf{E}_1(\mu_0) = \mathsf{E}[\mu(X, D = 0, M)|X, D = 1]$.

Table A8: The validation losses of nuisance functions in real data application to the COMPAS algorithm fairness.

| | DeepMed | Lasso | RF | GBM |
|---|---|---|---|---|
| $a(1|x,m)$ | 0.622 | 0.626 | 0.638 | 0.625 |
| $a(1|x)$ | 0.648 | 0.650 | 0.699 | 0.650 |
| $\mu(x,1,m)$ | 4.924 | 5.480 | 5.436 | 5.064 |
| $\mathsf{E}[\mu(X, D = 1, M)|X = x, D = 0]$ | 1.636 | 0.993 | 0.832 | 1.579 |
| $\mu(x,1)$ | 7.265 | 7.392 | 7.393 | 7.378 |
| $\mu(x,0,m)$ | 3.710 | 4.108 | 4.266 | 3.928 |
| $\mathsf{E}[\mu(X, D = 0, M)|X = x, D = 1]$ | 7.582 | 2.443 | 0.993 | 2.012 |
| $\mu(x,0)$ | 5.197 | 5.299 | 5.414 | 5.269 |

Table A9: Simulation Case 4 ($\alpha = 0.6$): The biases, empirical standard errors (SE) and root mean squared errors (RMSE) of the estimated average treatment effects and coverage probabilities (CP) of 95% confidence intervals. The simulation is based on 200 replicates.

| | Method | $p = 5$ | | | | $p = 20$ | | | | $p = 100$ | | | |
|---|---|---|---|---|---|---|---|---|---|---|---|---|---|
| | | Bias | SE | RMSE | CP | Bias | SE | RMSE | CP | Bias | SE | RMSE | CP |
| $\tau_{tot}$ | DeepMed | 0.027 | 0.047 | 0.054 | 0.880 | 0.035 | 0.047 | 0.059 | 0.880 | 0.062 | 0.051 | 0.080 | 0.740 |
| | Lasso | 0.353 | 0.087 | 0.364 | 0.010 | 0.335 | 0.082 | 0.345 | 0.020 | 0.343 | 0.087 | 0.354 | 0.020 |
| | RF | 0.004 | 0.051 | 0.051 | 0.995 | 0.080 | 0.058 | 0.099 | 0.780 | 0.202 | 0.065 | 0.212 | 0.095 |
| | GBM | 0.020 | 0.050 | 0.054 | 0.930 | 0.055 | 0.056 | 0.078 | 0.830 | 0.088 | 0.061 | 0.107 | 0.650 |
| | Oracle | -0.002 | 0.031 | 0.031 | 0.955 | -0.003 | 0.031 | 0.031 | 0.950 | 0.002 | 0.031 | 0.031 | 0.930 |
| $\tau_{NDE}(1)$ | DeepMed | 0.004 | 0.038 | 0.038 | 0.925 | -0.006 | 0.041 | 0.041 | 0.940 | 0.003 | 0.047 | 0.047 | 0.945 |
| | Lasso | 0.227 | 0.071 | 0.238 | 0.095 | 0.211 | 0.066 | 0.221 | 0.120 | 0.217 | 0.073 | 0.229 | 0.115 |
| | RF | 0.022 | 0.045 | 0.050 | 0.980 | 0.087 | 0.054 | 0.102 | 0.690 | 0.175 | 0.060 | 0.185 | 0.140 |
| | GBM | 0.017 | 0.051 | 0.054 | 0.905 | 0.044 | 0.057 | 0.072 | 0.845 | 0.073 | 0.062 | 0.096 | 0.750 |
| | Oracle | 0.000 | 0.022 | 0.022 | 0.965 | -0.003 | 0.022 | 0.022 | 0.920 | 0.002 | 0.022 | 0.022 | 0.940 |
| $\tau_{NDE}(0)$ | DeepMed | 0.005 | 0.034 | 0.034 | 0.945 | 0.005 | 0.038 | 0.038 | 0.960 | 0.014 | 0.047 | 0.049 | 0.925 |
| | Lasso | 0.229 | 0.069 | 0.239 | 0.090 | 0.213 | 0.066 | 0.223 | 0.115 | 0.217 | 0.073 | 0.229 | 0.125 |
| | RF | 0.032 | 0.047 | 0.057 | 0.960 | 0.081 | 0.051 | 0.096 | 0.730 | 0.158 | 0.061 | 0.169 | 0.210 |
| | GBM | 0.023 | 0.050 | 0.055 | 0.930 | 0.049 | 0.056 | 0.074 | 0.865 | 0.075 | 0.061 | 0.097 | 0.715 |
| | Oracle | 0.000 | 0.022 | 0.022 | 0.965 | -0.003 | 0.023 | 0.023 | 0.925 | 0.002 | 0.021 | 0.021 | 0.950 |
| $\tau_{NIE}(1)$ | DeepMed | 0.021 | 0.039 | 0.044 | 0.865 | 0.030 | 0.042 | 0.052 | 0.865 | 0.047 | 0.044 | 0.064 | 0.830 |
| | Lasso | 0.124 | 0.051 | 0.134 | 0.295 | 0.123 | 0.048 | 0.132 | 0.280 | 0.126 | 0.049 | 0.135 | 0.260 |
| | RF | -0.029 | 0.040 | 0.049 | 0.895 | -0.002 | 0.039 | 0.039 | 0.935 | 0.044 | 0.038 | 0.058 | 0.760 |
| | GBM | -0.003 | 0.042 | 0.042 | 0.935 | 0.006 | 0.045 | 0.045 | 0.905 | 0.013 | 0.039 | 0.041 | 0.935 |
| | Oracle | -0.002 | 0.024 | 0.024 | 0.940 | 0.000 | 0.023 | 0.023 | 0.930 | -0.001 | 0.023 | 0.023 | 0.950 |
| $\tau_{NIE}(0)$ | DeepMed | 0.023 | 0.040 | 0.046 | 0.875 | 0.041 | 0.041 | 0.058 | 0.815 | 0.059 | 0.047 | 0.075 | 0.730 |
| | Lasso | 0.126 | 0.052 | 0.136 | 0.270 | 0.124 | 0.048 | 0.133 | 0.250 | 0.126 | 0.049 | 0.135 | 0.250 |
| | RF | -0.019 | 0.042 | 0.046 | 0.935 | -0.007 | 0.035 | 0.036 | 0.955 | 0.027 | 0.036 | 0.045 | 0.885 |
| | GBM | 0.004 | 0.042 | 0.042 | 0.920 | 0.012 | 0.041 | 0.043 | 0.930 | 0.015 | 0.039 | 0.042 | 0.930 |
| | Oracle | -0.001 | 0.024 | 0.024 | 0.945 | 0.000 | 0.023 | 0.023 | 0.925 | -0.001 | 0.023 | 0.023 | 0.955 |

Table A10: Simulation Case 5: The biases, empirical standard errors (SE) and root mean squared errors (RMSE) of the estimated average treatment effects and coverage probabilities (CP) of 95% confidence intervals. The simulation is based on 200 replicates.

| | Method | Bias | SE | RMSE | CP |
|---|---|---|---|---|---|
| $\tau_{tot}$ | DeepMed | 0.225 | 0.032 | 0.227 | 0.000 |
| | Lasso | 0.369 | 0.032 | 0.370 | 0.000 |
| | RF | 0.277 | 0.035 | 0.279 | 0.000 |
| | GBM | 0.368 | 0.032 | 0.369 | 0.000 |
| | Oracle | -0.003 | 0.028 | 0.028 | 0.960 |
| $\tau_{\text{NDE}}(1)$ | DeepMed | 0.040 | 0.023 | 0.046 | 0.570 |
| | Lasso | 0.113 | 0.023 | 0.115 | 0.005 |
| | RF | 0.087 | 0.029 | 0.092 | 0.170 |
| | GBM | 0.114 | 0.024 | 0.116 | 0.005 |
| | Oracle | -0.001 | 0.020 | 0.020 | 0.965 |
| $\tau_{\text{NDE}}(0)$ | DeepMed | 0.044 | 0.023 | 0.050 | 0.470 |
| | Lasso | 0.112 | 0.023 | 0.114 | 0.005 |
| | RF | 0.083 | 0.029 | 0.088 | 0.200 |
| | GBM | 0.114 | 0.024 | 0.116 | 0.005 |
| | Oracle | -0.002 | 0.020 | 0.020 | 0.960 |
| $\tau_{\text{NIE}}(1)$ | DeepMed | 0.181 | 0.027 | 0.183 | 0.000 |
| | Lasso | 0.257 | 0.026 | 0.258 | 0.000 |
| | RF | 0.193 | 0.033 | 0.196 | 0.000 |
| | GBM | 0.255 | 0.027 | 0.256 | 0.000 |
| | Oracle | -0.001 | 0.021 | 0.021 | 0.955 |
| $\tau_{\text{NIE}}(0)$ | DeepMed | 0.184 | 0.028 | 0.186 | 0.000 |
| | Lasso | 0.256 | 0.027 | 0.257 | 0.000 |
| | RF | 0.190 | 0.034 | 0.193 | 0.000 |
| | GBM | 0.255 | 0.027 | 0.256 | 0.000 |
| | Oracle | -0.002 | 0.021 | 0.021 | 0.940 |

Table A11: The simulation results under Cases 4-5 for DeepMed with DNN weights trained by SGD. The biases, empirical standard errors (SE) and root mean squared errors (RMSE) of the estimated average treatment effects and coverage probabilities (CP) of 95% confidence intervals. The simulation is based on 200 replicates.

| | | | Bias | SE | RMSE | CP |
|---|---|---|---|---|---|---|
| Case 4 | $p = 5$ | $\tau_{tot}$ | 0.033 | 0.049 | 0.059 | 0.870 |
| | | $\tau_{\text{NDE}}(1)$ | 0.007 | 0.037 | 0.038 | 0.955 |
| | | $\tau_{\text{NDE}}(0)$ | 0.004 | 0.039 | 0.039 | 0.940 |
| | | $\tau_{\text{NIE}}(1)$ | 0.028 | 0.041 | 0.050 | 0.825 |
| | | $\tau_{\text{NIE}}(0)$ | 0.026 | 0.040 | 0.048 | 0.815 |
| | $p = 20$ | $\tau_{tot}$ | 0.060 | 0.050 | 0.078 | 0.770 |
| | | $\tau_{\text{NDE}}(1)$ | 0.006 | 0.045 | 0.045 | 0.925 |
| | | $\tau_{\text{NDE}}(0)$ | 0.011 | 0.044 | 0.045 | 0.910 |
| | | $\tau_{\text{NIE}}(1)$ | 0.049 | 0.043 | 0.065 | 0.710 |
| | | $\tau_{\text{NIE}}(0)$ | 0.054 | 0.048 | 0.072 | 0.715 |
| | $p = 100$ | $\tau_{tot}$ | 0.073 | 0.050 | 0.088 | 0.655 |
| | | $\tau_{\text{NDE}}(1)$ | 0.026 | 0.059 | 0.064 | 0.835 |
| | | $\tau_{\text{NDE}}(0)$ | 0.027 | 0.055 | 0.061 | 0.865 |
| | | $\tau_{\text{NIE}}(1)$ | 0.046 | 0.051 | 0.069 | 0.775 |
| | | $\tau_{\text{NIE}}(0)$ | 0.047 | 0.059 | 0.075 | 0.745 |
| Case 5 | | $\tau_{tot}$ | 0.229 | 0.034 | 0.232 | 0.000 |
| | | $\tau_{\text{NDE}}(1)$ | 0.043 | 0.024 | 0.049 | 0.495 |
| | | $\tau_{\text{NDE}}(0)$ | 0.040 | 0.024 | 0.047 | 0.590 |
| | | $\tau_{\text{NIE}}(1)$ | 0.190 | 0.027 | 0.192 | 0.000 |
| | | $\tau_{\text{NIE}}(0)$ | 0.186 | 0.027 | 0.188 | 0.000 |

Table A12: Real data application to income fairness. The estimated NDE/NIE of gender ($D$) on income ($Y$) with occupation ($M$) as the mediator.

| Method | Effect | Estimate | SE | P value |
|---|---|---|---|---|
| DeepMed | $\tau_{tot}$ | 0.155 | 0.004 | $< 10^{-16}$ |
| | $\tau_{\mathsf{NDE}}(1)$ | 0.161 | 0.007 | $< 10^{-16}$ |
| | $\tau_{\mathsf{NDE}}(0)$ | 0.148 | 0.004 | $< 10^{-16}$ |
| | $\tau_{\mathsf{NIE}}(1)$ | 0.007 | 0.002 | 0.003 |
| | $\tau_{\mathsf{NIE}}(0)$ | -0.005 | 0.005 | 0.343 |
| Lasso | $\tau_{tot}$ | 0.171 | 0.004 | $< 10^{-16}$ |
| | $\tau_{\mathsf{NDE}}(1)$ | 0.165 | 0.006 | $< 10^{-16}$ |
| | $\tau_{\mathsf{NDE}}(0)$ | 0.155 | 0.004 | $< 10^{-16}$ |
| | $\tau_{\mathsf{NIE}}(1)$ | 0.016 | 0.002 | $3 \times 10^{-11}$ |
| | $\tau_{\mathsf{NIE}}(0)$ | 0.006 | 0.004 | 0.160 |
| RF | $\tau_{tot}$ | 0.092 | 0.005 | $< 10^{-16}$ |
| | $\tau_{\mathsf{NDE}}(1)$ | 0.153 | 0.003 | $< 10^{-16}$ |
| | $\tau_{\mathsf{NDE}}(0)$ | 0.114 | 0.006 | $< 10^{-16}$ |
| | $\tau_{\mathsf{NIE}}(1)$ | -0.022 | 0.003 | $5 \times 10^{-12}$ |
| | $\tau_{\mathsf{NIE}}(0)$ | -0.060 | 0.003 | $< 10^{-16}$ |
| GBM | $\tau_{tot}$ | 0.157 | 0.004 | $< 10^{-16}$ |
| | $\tau_{\mathsf{NDE}}(1)$ | 0.152 | 0.006 | $< 10^{-16}$ |
| | $\tau_{\mathsf{NDE}}(0)$ | 0.146 | 0.004 | $< 10^{-16}$ |
| | $\tau_{\mathsf{NIE}}(1)$ | 0.011 | 0.002 | $5 \times 10^{-6}$ |
| | $\tau_{\mathsf{NIE}}(0)$ | 0.005 | 0.004 | 0.247 |

Table A13: The validation losses of nuisance functions in real data application to income fairness.

| | DeepMed | Lasso | RF | GBM |
|---|---|---|---|---|
| $a(1\|x,m)$ | 0.501 | 0.516 | 0.560 | 0.502 |
| $a(1\|x)$ | 0.600 | 0.612 | 0.631 | 0.600 |
| $\mu(x,1,m)$ | 0.465 | 0.493 | 0.681 | 0.467 |
| $\mathsf{E}[\mu(X, D = 1, M)\|X = x, D = 0]$ | 0.010 | 0.011 | 0.024 | 0.007 |
| $\mu(x,1)$ | 0.479 | 0.510 | 1.040 | 0.480 |
| $\mu(x,0,m)$ | 0.285 | 0.300 | 0.478 | 0.287 |
| $\mathsf{E}[\mu(X, D = 0, M)\|X = x, D = 1]$ | 0.003 | 0.005 | 0.002 | 0.002 |
| $\mu(x,0)$ | 0.288 | 0.306 | 0.711 | 0.291 |