# OpenReview forum: "DeepMed: Semiparametric Causal Mediation Analysis with Debiased Deep Learning"
_NeurIPS.cc/2022/Conference — NeurIPS 2022 Accept_

### Official Review · Reviewer_NkWa · 2022-06-16

**Rating:** 6
**Confidence:** 3
**Soundness:** 3 good
**Presentation:** 2 fair
**Contribution:** 3 good

**Summary:**

This paper proposes a semi-parametric method for estimating direct and indirect effects from the observational data with deep neural networks.

**Questions:**

I think the paper doesn't raise too many questions

**Limitations:**

- somewhat unclear what the limitations are
- societal impact has been addressed

**Strengths And Weaknesses:**

- The paper is somewhat hard to follow with a lot of Mathematical notation that would benefit from a bit better explanation
- Quality of theoretical and experimental results appears sound - even though I did not go through any derivations
- The reviewer is unclear on the level of novelty/originality it appears to be ok but can't say for certain

---

> ### Author Response · Authors · 2022-07-31
> **thank you for your review; our response**
>
> First off, we thank Reviewer NkWa for the overall positive feedback on our paper. We highly appreciate your time and effort!
>
> The novelty of our paper is mainly two-fold:
>
> (1) We obtained state-of-the-art theoretical properties of DNN-based causal effect estimation and improved the results of a high-impact paper by Farrell et al. 2021 [1] on several fronts, including allowing dense neural nets architecture (more commonly used than sparse neural nets) and leveraging the potential low-dimensional structure of the true nuisance functions, both of which are important in practice;
>
> (2) We designed nontrivial synthetic experiments that reveal the potential adverse effect on causal effect estimation due to the implicit regularization by the gradient-based training algorithms such as SGD or adam. We believe the above two contributions are important to the causal-AI/ML community because both points are on the path of making the theoretical results closer to guiding practice.
>
> Your comments on the writing styles are also well received. In the revision, we tried to add more explanations for the math-heavy parts, without significantly increasing the length or changing the overall structure of our paper. We hope that our revision improves the presentation. In particular, we made the following changes:
>
> 1) We added more explanations on the ignorability assumption at the bottom of page 2 and in Appendix A, as suggested by Reviewer gGUT.
>
> 2) We added more backgrounds on Holder spaces on the bottom of page 5 of the main text.
>
> 3) Before Lemma 5 (page 6), we added words to explain the implication of Lemma 5 and emphasized early in the text what is the main difference between the results in Farrell et al. 2021 [1] and ours.
>
> 4) For Theorem 6 (page 6), we further explained what the in-probability convergence of $\hat{\sigma}^{2}$ means.
>
> We are more than willing to accommodate more changes to improve the presentation of our paper if there are more concrete suggestions.
>
> Finally, we did discuss the limitations of our paper in Conclusions and Discussions at the end of page 9, pointing towards at least two future research directions: 1) taking into account the implicit regularization effect of the training process when deriving the theoretical properties of DNN-based mediation analysis and 2) incorporating techniques for dealing with unmeasured confounding into DNN-based mediation analysis, such as instrumental variables and the proximal causal learning framework.
>
> We hope that our revision and response to your comments have addressed your concerns and now you also share the same enthusiasm for this work as we do!
>
> __References:__
>
> [1] Max H Farrell, Tengyuan Liang, and Sanjog Misra. Deep neural networks for estimation and inference. Econometrica, 89(1):181–213, 2021.

---

### Official Review · Reviewer_UV1a · 2022-07-09

**Rating:** 7
**Confidence:** 1
**Soundness:** 3 good
**Presentation:** 4 excellent
**Contribution:** 3 good

**Summary:**

This work proposes DeepMed, which reduces bias in semiparametric mediation analysis using the power of neural networks. It relaxes sparsity assumptions of the prior work on the theoretical side which gives more flexibility and expressivity to neural networks and validates it with synthetic and real-world data experiments.

**Questions:**

In remark 10, the authors conjecture that the reason DeepMed is not semiparametric efficient in cases 4 and 5 is due to implicit regularization of Adam. It would be interesting to test this conjecture by running these experiments with normal SGD as the optimizer, and checking if DeepMed becomes efficient.

**Limitations:**

yes

**Strengths And Weaknesses:**

The Paper is well written.

---

> ### Author Response · Authors · 2022-07-31
> **thank you for your review; our response**
>
> We thank Reviewer UV1a for the positive feedback and insightful comment on our paper. We highly appreciate your time and effort!
>
> Reviewer UV1a raised an important question on whether using SGD instead of adam can alleviate the effect of implicit regularization by adam on mediation analysis. We apologize that our original Remark 10 did not make this point clear. Our conjecture actually applies to all gradient-based algorithms, including SGD and adam. To demonstrate this point, we conducted extra experiments (see page 18 and Table A11 on page 26 of the revised online Appendix.pdf) on Cases 4, 5 by training DNNs using SGD instead of adam, as Reviewer UV1a suggested. As we expect, SGD does not lead to semiparametric efficient NDE, NIE, and total effect estimators either. This is reassuring that the natural and necessary next step for theoretical development is to fully incorporate the training process into the statistical properties of causal effect estimation. Without our nontrivial synthetic experiments, it would not have been so clear that this is the case. We have also revised Remark 10 accordingly to make the above point clear.
>
> We hope that our response has addressed your question and thank you again for your positive evaluation of our paper!

---

> > ### Author Response · Authors · 2022-08-07
> > **thank you again for your time and thoughtful comments**
> >
> > Dear reviewer UV1a,
> >
> > We would like to thank you again for your time and thoughtful comments about our work. We are wondering whether our response has sufficiently addressed your concerns. We would be happy to do our best to address further questions or comments. Please let us know if you still have any unclear parts of our work.
> >
> > Thank you very much!
> >
> > Best regards,
> >
> > Authors of paper 5618

---

### Official Review · Reviewer_7FFK · 2022-07-11

**Rating:** 5
**Confidence:** 1
**Soundness:** 3 good
**Presentation:** 3 good
**Contribution:** 3 good

**Summary:**

This work proposes DeepMed, a semiparametric causal medication analysis framework, which focuses on natural direct/indirect effects (NDE/NIE) in the causal analysis domain. By leveraging the second-order bias property of the multiply-robust estimators of NDE/NIE, DeepMed is able to outperform baseline methods in both simulated datasets and real datasets.

**Questions:**

N/A

**Strengths And Weaknesses:**

Strengths:
The authors provide extensive theoretical analysis as to the semiparametric multiply-robust estimators of NDE/NIE, as well as the statistical properties of DeepMed

Weaknesses:
The authors rely on synthetic experiments to provide in-depth analysis of how DeepMed is able to outperform baselines as well as perform better causal mediation analysis.

---

> ### Author Response · Authors · 2022-07-31
> **thank you for your review; our response**
>
> First, we thank Reviewer 7FFK for the overall positive feedback on our paper. We highly appreciate your time and effort!
>
> Reviewer 7FFK commented that the main weakness of our paper is evaluating DeepMed based only on synthetic experiments. From the practical perspective, we definitely agree with Reviewer 7FFK that it is ideal to evaluate a new method in as many real datasets as possible. But for causal inference tasks, this is not easy because we rarely know the ground truth, and therefore carefully designed synthetic experiments are still critically important for methods evaluation.
>
> Furthermore, as in our response to Reviewer NkWa, the synthetic experiments designed in our paper are __highly nontrivial__. Such experiments help reveal the effect on causal effect estimation due to the implicit regularization by the gradient-based training algorithms such as SGD or adam. This issue has not been fully appreciated by the causal inference community; for example, for the papers on DNN causal inference that we cited in our paper, none of them discussed this issue, even though implicit regularization/bias has been observed in many settings. We believe it is extremely important to make practitioners in causal inference aware of this issue, considering that causal inference tasks are often involved in high-stake applications.
>
> We hope that our response has addressed your concerns and now you also share the same enthusiasm for this work as we do!

---

> > ### Author Response · Authors · 2022-08-07
> > **thank you again for your time and thoughtful comments**
> >
> > Dear reviewer 7FFK,
> >
> > We would like to thank you again for your time and thoughtful comments about our work. We are wondering whether our response has sufficiently addressed your concerns. We would be happy to do our best to address further questions or comments. Please let us know if you still have any unclear parts of our work.
> >
> > Thank you very much!
> >
> > Best regards,
> >
> > Authors of paper 5618

---

> > > ### Comment · Reviewer_7FFK · 2022-08-09
> > > **Thank you for your response**
> > >
> > > I appreciate the authors' explanation, but essentially nothing has changed in terms of paper content, and hence I will keep my original score.

---

### Official Review · Reviewer_gGUT · 2022-07-22

**Rating:** 6
**Confidence:** 4
**Soundness:** 3 good
**Presentation:** 3 good
**Contribution:** 2 fair

**Summary:**

The authors propose a new method called DeepMed for performing semiparametric mediation analysis with DNNs.


**Questions:**


- In remark 3, authors have suggested to use a(d | x, m) and a(d | x) to avoid curse of dimensionality in fitting f(m | x, d) if m is multidimensional and or continuous. However, this alternative solution can lead to new complications such as incompatibility in posing models for a(d | x, m) and a(d | x) while having a coherent joint over X, D, M, Y. That would be great if authors could comment on this and possibly provide remedies to resolve the issue.

- In section 2.2., It would be more clear if the “cross-world” ignobility assumption is written down more explicitly, i.e.,  Y(d, m) \indep M(d’) \mid X

- The identifying functional on top of page 3, should have an integration over dPx as well.

- Why is X defined as {0, 1}^p in Section 2.1? The arguments don't seem to rely on restricted state space on X.

**Strengths And Weaknesses:**

Their main theoretical contribution is using the multiply-robust behavior of IF-based estimators for (in)direct effect in order to relax the sparsity constraints in training nuisance functions with DNN architectures.

However, this work https://academic.oup.com/ectj/article/25/2/277/6517682 (and possibly others) discusses most of what hacve been proposed in this draft regarding the second order bias and properties of the estimator within the sample splitting scheme (theorem 1).

I’m less familiar with the DNN literature, and can’t comment on the novelty of the theoretical claims specific to using DNNs to fit the nuisances. But I find a full-fledged procedure to fit nuisances via DNNs to compute mediated effects worthwhile and not immediately straightforward.

I also think the authors have done a good job with the synthetic experiments.

---

> ### Author Response · Authors · 2022-07-31
> **thank you for your review; our response**
>
> We thank Reviewer gGUT for the positive feedback and detailed comments on our paper. We highly appreciate your time and effort!
>
> First, we agree that the second-order bias is a direct consequence of the first-order influence function of NDE/NIE originally derived in Tchetgen Tchetgen and Shpitser 2012 [1]. In our original paper, we did cite both [1] and the paper mentioned by Reviewer gGUT, Farbmacher et al. 2022 [2] (now published in The Econometrics Journal). To highlight this point, we now changed the name "Theorem 1" to "Proposition 1" and added more background on its statement afterward. __Nonetheless, as also acknowledged by Reviewer gGUT, the focus of our paper is on the theoretical properties and empirical performance of mediation analysis when DNNs are used to estimate the nuisance functions.__ We believe this is an important problem as it is highly likely that DNNs will be used by practitioners more frequently in mediation analysis. This is also not a straightforward problem. Our theoretical result is state-of-the-art, improving Farrell et al. 2021 [3] on several fronts. We summarized our contributions and their importance at the end of the Introduction section.
>
> Second, let us respond to your four questions one by one:
>
> (1) This is a very insightful comment! This could be an issue if one models $a (d | x, m)$ and $a (d | x)$ separately by positing parametric models. In this paper, however, we take the semiparametric viewpoint by modeling the nuisance functions, including $a (d | x, m)$ and $a (d | x)$, nonparametrically (using overparameterized DNNs) and viewing NDE/NIE as the low-dimensional parameter of actual scientific interests. As long as the nuisance functions are estimated at sufficiently fast rates (in $L_{2}$ sense in our paper), the estimator of the causal parameter should be accurate. A similar perspective has been taken by other papers from the semiparametric literature as well, e.g. [4, 5], where they even allowed the propensity score (probability) estimates to lie outside $[0, 1]$. This is indeed one of the advantages of the semiparametric framework, which does not put too much emphasis on modeling the joint distribution of the observed data $(X, A, M, Y)$. We added more discussion on this issue in Remark 3, and mostly in Appendix G of the revised paper due to space constraints. But if Reviewer gGUT prefers the discussions to be in the main text, we are more than happy to make further changes.
>
> (2) We have now revised the ignorability assumptions part accordingly and added Appendix A to further comment on these assumptions. We totally agree that this version reads much better and thank you for this great suggestion!
>
> (3) We have now corrected this typo. Thank you again for reading our paper so carefully.
>
> (4) The state space of $X$ is actually $[0, 1]^{p}$ instead of the Boolean hypercube {0, 1}$^{p}$. In fact, as long as $X$ is compactly supported in $\mathbb{R}^{p}$ (commonly assumed in nonparametric statistic and deep learning literature [6, 7]), our theoretical results should go through. We now briefly comment on this issue right after we mention the state space of $X$ in Section 2.1.
>
> Finally, we are grateful for your positive comment on our synthetic experiments. In our own humble opinion, we do believe that such synthetic experiments should be done in any other works evaluating their theoretical claims related to DNN-based causal effect estimation, if they take the nonparametric paradigm. We hope that our package can be useful to researchers in causal inference, deep learning, and machine learning in general.
>
> Hopefully, our response has fully addressed your concerns about our paper and now you also share the same enthusiasm for this paper as we do.
>
> __References:__
>
> [1] Eric J Tchetgen Tchetgen and Ilya Shpitser. Semiparametric theory for causal mediation analysis: Efficiency bounds, multiple robustness and sensitivity analysis. The Annals of Statistics, 40(3): 1816–1845, 2012.
>
> [2] Helmut Farbmacher, Martin Huber, Lukas Laffers, Henrika Langen, and Martin Spindler. Causal mediation analysis with double machine learning. The Econometrics Journal, 25(2):277–300, 2022.
>
> [3] Max H Farrell, Tengyuan Liang, and Sanjog Misra. Deep neural networks for estimation and inference. Econometrica, 89(1):181–213, 2021.
>
> [4] Whitney K Newey, and James M Robins. Cross-fitting and fast remainder rates for semiparametric estimation. arXiv preprint arXiv:1801.09138, 2018.
>
> [5] Jelena Bradic, Victor Chernozhukov, Whitney K Newey, and Yinchu Zhu. Minimax Semiparametric Learning With Approximate Sparsity. arXiv preprint arXiv:1912.12213, 2019.
>
> [6] Evarist Gine, and Richard Nickl. Mathematical foundations of infinite-dimensional statistical models. Cambridge University Press, volume 40, 2016.
>
> [7] Johannes Schmidt-Hieber. Nonparametric regression using deep neural networks with ReLU activation function. Annals of Statistics, 48(4):1875–1897, 2020.

---

> > ### Author Response · Authors · 2022-08-07
> > **thank you again for your time and thoughtful comments**
> >
> > Dear reviewer gGUT,
> >
> > We would like to thank you again for your time and thoughtful comments about our work. We are wondering whether our response has sufficiently addressed your concerns. We would be happy to do our best to address further questions or comments. Please let us know if you still have any unclear parts of our work.
> >
> > Thank you very much!
> >
> > Best regards,
> >
> > Authors of paper 5618

---

### Author Response · Authors · 2022-08-08
**response**

Dear reviewers,

Did we address all your questions? Do you have any further questions? The time window to respond for us is closing tomorrow. We are more than willing to address any issues that remain.

Best Regards,

Authors of Paper 5618

---

### Meta-Review · Area_Chair_Sens · 2022-08-27

**Recommendation:** Accept
**Confidence:** Certain

**Metareview:**

The decision is to accept the paper.

The paper provides a theoretical treatment of using DNN's for nuisance function estimation in multiply-robust mediation analysis. The theory developed here is DNN-specific, and novel synthetic experiments that better test the ability of nuisance functions to adapt to complex ground-truth functions are proposed. A proof-of-concept demonstration on fairness-oriented mediation analysis is also provided.

While the subject matter is dense, the paper builds on a well-established line of work, and makes solid theoretical and evaluation-design contributions. I agree with the authors that in mediation analysis, synthetic evaluation is in many ways the best we can do, especially for evaluating the theoretical claims in this paper.

One suggestion I have from looking over the paper: In the experiments section, I hope the authors can make clearer what the empirical basis is for judging a method to be semi-parametric efficient in their synthetic experiments (e.g., the authors note that DeepMed is not semiparamertric efficient in some of the synthetic settings---which is fine---but don't explain in the main text how such a judgment was made). I suspect this comes from comparing performance to the estimate that used oracle nuisance functions under appropriately large sample size, but this should be stated explicitly, especially for the benefit of others in the community who might want to extend this work using the evaluation framework developed in this paper.

**Award:**

No

---

### Decision · Program_Chairs · 2022-09-14

Accept